# VoGE: A Differentiable Volume Renderer using Gaussian Ellipsoids for Analysis-by-Synthesis

Angtian Wang[1], Peng Wang[2], Jian Sun[2], Adam Kortylewski[3], and Alan Yuille[1]

[1] Johns Hopkins University
[2] ByteDance Inc.
[3] Max Planck Institute for Informatics

## ABSTRACT

The Gaussian reconstruction kernels have been proposed by Westover (1990) and studied by the computer graphics community back in the 90s, which gives an alternative representation of object 3D geometry from meshes and point clouds. On the other hand, current state-of-the-art (SoTA) differentiable renderers, Liu et al. (2019), use rasterization to collect triangles or points on each image pixel and blend them based on the viewing distance. In this paper, we propose VoGE, which utilizes the volumetric Gaussian reconstruction kernels as geometric primitives. The VoGE rendering pipeline uses ray tracing to capture the nearest primitives and blends them as mixtures based on their volume density distributions along the rays. To efficiently render via VoGE, we propose an approximate close-form solution for the volume density aggregation and a coarse-to-fine rendering strategy. Finally, we provide a CUDA implementation of VoGE, which enables real-time level rendering with a competitive rendering speed in comparison to PyTorch3D. Quantitative and qualitative experiment results show VoGE outperforms SoTA counterparts when applied to various vision tasks, *e.g.*, object pose estimation, shape/texture fitting, and occlusion reasoning. The code is available: https://github.com/Angtian/VoGE.

## 1 INTRODUCTIONS

Recently, the integration of deep learning and computer graphics has achieved significant advances in lots of computer vision tasks, *e.g.*, pose estimation Wang et al. (2020a), 3D reconstruction Zhang et al. (2021), and texture estimation Bhattad et al. (2021). Although the rendering quality of has significant improved over decades of development of computer graphics, the differentiability of the rendering process still remains to be explored and improved. Specifically, differentiable renderers compute the gradients w.r.t. the image formation process, and hence enable to broadcast cues from 2D images towards the parameters of computer graphics models, such as the camera parameters, and object geometries and textures. Such an ability is also essential when combining graphics models with deep neural networks. In this work, we focus on developing a differentiable renderer using explicit object representations, *i.e.* Gaussian reconstruction kernels, which can be either used separately for image generation or for serving as 3D aware neural network layers.

The traditional rendering process typically involves a naive rasterization Kato et al. (2018), which projects geometric primitives onto the image plane and only captures the nearest primitive for each pixel. However, this process eliminates the cues from the occluded primitives and blocks gradients toward them. Also the rasterization process introduces a limitation for differentiable rendering, that rasterization assumes primitives do not overlap with each other and are ordered front to back along the viewing direction Zwicker et al. (2001). Such assumption raise a paradox that during gradient based optimization, the primitives are necessary to overlap with each other when they change the order along viewing direction. Liu et al. (2019) provide a naive solution that tracks a set of nearest primitives for each image pixel, and blending them based on the viewing distance. However, such

Figure 1: VoGE conducts ray tracing volume densities. Given the Gaussian Ellipsoids, *i.e.* a set of ellipsoidal 3D Gaussian reconstruction kernels, VoGE first samples rays $r(t)$. And along each ray, VoGE traces the density distribution of each ellipsoid $\rho_k(r(t))$ respectively. Then occupancy $T(r(t))$ is accumulated via density aggregation along the ray. The observation of each Gaussian ellipsoid kernels $W_k$ is computed via integral of reweighted per-kernel volume density $W_k(r(t))$. Finally, VoGE synthesizes the image using the computed $W_k$ on each pixel to interpolate per kernel attributes. In practice, the density aggregation is bootstrapped via approximate close-form solutions.

approach introduces an ambiguity that, for example, there is a red object floating in front of a large blue object laying as background. Using the distance based blending method, the cues of the second object will change when moving the red one from far to near, especially when the red object are near the blue one, which will give a unrealistic purple blending color. In order to resolve the ambiguity, we record the volume density distributions instead of simply recording the viewing distance, since such distributions provide cues on occlusion and interaction of primitives when they overlapped.

Recently, Mildenhall et al. (2020); Schwarz et al. (2020) show the power of volume rendering with high-quality occlusion reasoning and differentiability, which benefits from the usage of ray tracing volume densities introduced by Kajiya & Von Herzen (1984). However, the rendering process in those works relies on implicit object representations which limits the modifiability and interpretability. Back in the 90s, Westover (1990); Zwicker et al. (2001) develop the splatting method, which reconstruct objects using volumetric Gaussian kernels and renders based on a simplification of the ray tracing volume densities method. Unfortunately, splatting methods were designed for graphics rendering without considering the differentiability and approximate the ray tracing volume densities using rasterization. Inspired by both approaches, Zwicker et al. (2001); Liu et al. (2019), we propose VoGE using 3D Gaussians kernels to represent objects, which give soften boundary of primitives. Specifically, VoGE traces primitives along viewing rays as a density function, which gives a probability of observation along the viewing direction for the blending process.

In VoGE rendering pipeline, the ray tracing method is designed to replace rasterization, and a better blending function is developed based on integral of traced volume densities functions. As Figure 1 shows, VoGE uses a set of Gaussian ellipsoids to reconstruct the object in 3D space. Each Gaussian ellipsoid is indicated with a center location $\mathbf{M}$, and a spatial variance $\mathbf{\Sigma}$. During rendering, we first sample viewing rays by the camera configuration. We trace the volume density of each ellipsoid as a function of distance along the ray respectively on each ray, and compute the occupancy along the ray via an integral of the volume density and reweight the contribution of each ellipsoid. Finally, we interpolate the attribute of each reconstruction kernel with the kernel-to-pixel weights into an image. In practice, we propose an approximate close-form solution, which avoids computational heavy operation in the density aggregation without, *e.g.*, integral, cumulative sum. Benefited from advanced differentiability, VoGE obtains both high performance and speed on various vision tasks.

In summary, the contribution of VoGE includes:

1. A ray tracing method that traces each component along viewing ray as density functions. VoGE ray tracer is a replacement for rasterization.

2. A blending function based on integral of the density functions along viewing rays that reasons occlusion between primitives, which provides differentiability toward both visible and invisible primitives.

3. A differentiable CUDA implementation with real-time level rendering speed. VoGE can be easily inserted into neural networks via our PyTorch APIs.

4. Exceptional performance on various vision tasks. Quantitative results demonstrate that VoGE significantly outperforms concurrent state-of-the-art differentiable renderer on in-wild object pose estimation tasks.

Table 1: Comparison with state-of-the-art differentiable renderers. Similar to NeRF but different from previous graphics renderers, VoGE uses ray tracing to record volume densities on each ray for each ellipsoid, and blends them with transmittance computed via volume densities.

| Method | Representation | Primitives | Visibility Algorithm | Blending |
|---|---|---|---|---|
| Kato et al. (2018) | explicit | mesh | rasterization | none |
| Liu et al. (2019) | explicit | mesh | rasterization | distance |
| Ravi et al. (2020) | explicit | mesh/points | rasterization | distance |
| Yifan et al. (2019) | explicit | 2D Gaussian | rasterization | distance |
| Lassner & Zollhofer (2021) | explicit | sphere | rasterization | distance |
| VoGE (ours) | explicit | 3D Gaussian | ray tracing | transmittance |
| Mildenhall et al. (2020) | implicit | — | ray tracing | transmittance |

## 2 RELATED WORKS

**Volume Rendering.** In the 1980s, Blinn (1982) introduces the volume density representation, which simulates the physical process of light interacting with matter. Kajiya & Von Herzen (1984) develop the ray tracing volume density aggregation algorithm, which renders the volume density via light scattering equations. However, obtaining the contiguous volume density function is infeasible in practice. Currently Mildenhall et al. (2020); Niemeyer et al. (2020); Mescheder et al. (2019); Genova et al. (2020); Zhang et al. (2019); Vicini et al. (2022), use implicit functions, *e.g.*, neural networks, as object representations. Though those implicit representations give a satisfying performance, such representations are lacking interpretability and modifiability, which may limit their usage in analysis tasks. In this work, we provide a solution that utilizes explicit representation while rendering with the ray tracing volume density aggregation.

**Kernel Reconstruction of 3D Volume.** Westover (1990) introduces the volume reconstruction kernel, which decomposes a 3D volume into a sum of homogeneous primitives. Zwicker et al. (2001) introduces the elliptical Gaussian kernel and show such reconstruction gives satisfied shape approximation. However, both approaches conduct non-differentiable rendering and use rasterization to approximate the ray tracing process.

**Differentiable Renderer using Graphics.** Graphics renderers use explicit object representations, which represent objects as a set of geometry primitives. As Table 1 shows, concurrent differentiable graphics renderers use rasterization to determine visibility of primitives. In order to compute gradients across primitives boundaries, Loper & Black (2014); Kato et al. (2018); Liu et al. (2017) manually create the gradients while Liu et al. (2019); Yifan et al. (2019); Li et al. (2018); Nimier-David et al. (2019); Laine et al. (2020) use primitives with soft boundaries to allow gradients flow. Whereas to differentiate toward those occluded primitives, current differentiable renders, Liu et al. (2019); Yifan et al. (2019); Lassner & Zollhofer (2021), aggregate tracked primitives via viewing distance. However, all existing graphics renderers ignore the density distributions when conducting aggregation, which will introduce confusion while limiting differentiability.

**Renderer for Deep Neural Features.** Recent works demonstrate exceptional performance for rendering deep neural features. Specifically, for object pose estimation, Wang et al. (2020a); Iwase et al. (2021), demonstrate rendering on deep neural features benefits the optimization process in render-and-compare. Niemeyer & Geiger (2021) show rendering deep neural features also helps image generation tasks. In our work, we show VoGE benefits rendering using deep neural features via a better reconstruction of the spatial distribution of deep neural features.

## 3 VOLUME RENDERER FOR GAUSSIAN ELLIPSOIDS

In this section, we describe VoGE rendering pipeline that renders 3D Gaussians Ellipsoids into images under a certain camera configuration. Section 3.1 introduces the volume rendering. Section 3.2 describes the kernel reconstruction of the 3D volume using Gaussian ellipsoids. In Section 3.3, we propose the rendering pipeline for Gaussian ellipsoids via an approximate closed-form solution of ray tracing volume densities. Section 3.4 discusses the integration of VoGE with deep neural networks.

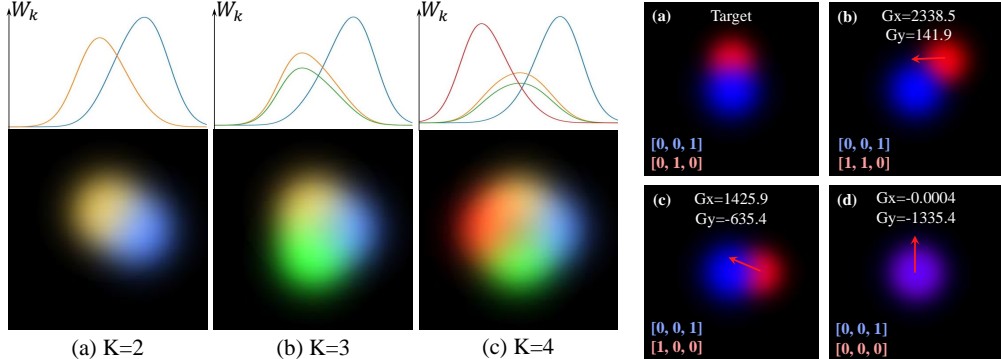

Figure 2: Rendering with increasing numbers of Gaussian Ellipsoids. Top: the kernel-to-pixel weight along the median row on the image, the colors demonstrate each corresponded Gaussian ellipsoids. Bottom: the rendered RGB image. Note VoGE resolves occlusion naturally in a contiguous way.

Figure 3: Computing gradient of $\mathbf{M}$ when rendering two ellipsoids. The colored numbers below indicate the $\mathbf{M}$ of each ellipsoids. The red arrow and $G_x, G_y$ show the $\frac{\partial(\mathbf{I}-\hat{\mathbf{I}})^2}{\partial\mathbf{M_{red}}}$.

### 3.1 VOLUME RENDERING

Different from the surface-based shape representations, in volume rendering, objects are represented using continuous volume density functions. Specifically, for each point in the volume, we have a corresponded density $\rho(x, y, z)$ with emitted color $c(x, y, z) = (r, g, b)$, where $(x, y, z)$ denotes the location of the point in the 3D space. Kajiya & Von Herzen (1984) propose using the light scattering equation during volume density, which provides a mechanism to compute the observed color $C(\mathbf{r})$ along a ray $\mathbf{r}(t) = (x(t), y(t), z(t))$:

$$C(\mathbf{r}) = \int_{t_n}^{t_f} T(t)\rho(\mathbf{r}(t))\mathbf{c}(\mathbf{r}(t))dt, \text{where } T(t) = \exp\left(-\tau \int_{t_n}^{t} \rho(\mathbf{r}(s))ds\right) \quad (1)$$

where $\tau$ is a coefficient that determines the rate of absorption, $t_n$ and $t_f$ denotes the near and far bound alone the ray, $T(t)$ is the transmittance.

### 3.2 GAUSSIAN ELLIPSOID RECONSTRUCTION KERNEL

Due to the difficulty of obtaining contiguous function of the volume density and enormous computation cost when calculating the integral, Westover (1990) introduces kernel reconstruction to conduct volume rendering in a computationally efficient way. The reconstruction decomposes the contiguous volume into a set of homogeneous kernels, while each kernel can be described with a simple density function. We use volume ellipsoidal Gaussians as the reconstruction kernels. Specifically, we reconstruct the volume with a sum of ellipsoidal Gaussians:

$$\rho(\mathbf{X}) = \sum_{k=1}^{K} \frac{1}{\sqrt{2\pi \cdot ||\mathbf{\Sigma}_k||_2}} e^{-\frac{1}{2}(\mathbf{X}-\mathbf{M}_k)^T \cdot \mathbf{\Sigma}_k^{-1} \cdot (\mathbf{X}-\mathbf{M}_k)} \quad (2)$$

where $K$ is the total number of Gaussian kernels, $\mathbf{X} = (x, y, z)$ is an arbitrary location in the 3D space. The $\mathbf{M}_k$, a $3 \times 1$ vector, is the center of $k$-th ellipsoidal Gaussians kernel. Whereas the $\mathbf{\Sigma}_k$ is a $3 \times 3$ spatial variance matrix, which controls the direction, size and shape of $k$-th kernel. Also, following Zwicker et al. (2001), we assume that the emitted color is approximately constant inside each reconstruction kernel $\mathbf{c}(\mathbf{r}(t)) = \mathbf{c}_k$.

The **VoGE mesh converter** creates Gaussian ellipsoids from a mesh. Specifically, we create Gaussians centered at all vertices' locations of the mesh. First, we compute a sphere-type Gaussians with same $\sigma_k$ on each direction, via average distance $\hat{l}$ from the center vertex to its connected neighbors, $\sigma_k = \frac{\hat{l}^2}{4 \cdot \log(1/\zeta)}$ where $\zeta$ is a parameter controls the Gaussians size. Then, we flatten the sphere-type Gaussians into ellipsoids with a flatten rate. Finally, for each Gaussian, we compute a rotation matrix via the mesh surface normal direction of the corresponded mesh vertex. We dot the rotation matrix onto the $\Sigma_k$ to make the Gaussians flatten along the surface.

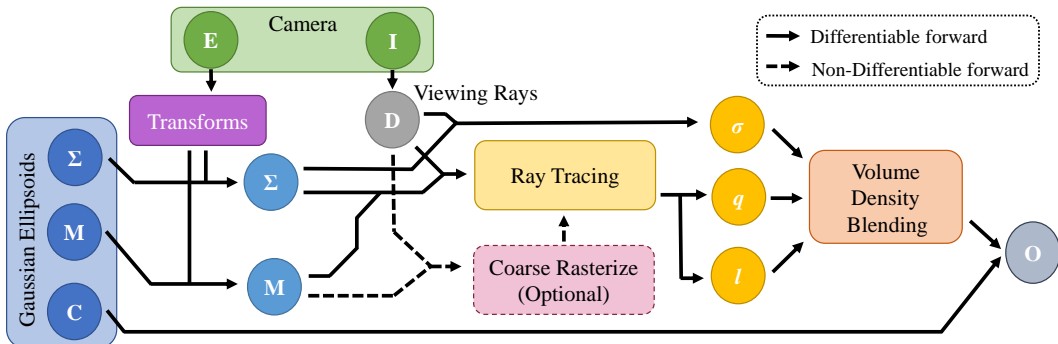

Figure 4: The forward process for VoGE rendering. The camera is described with the extrinsic matrix $\mathbf{E}$ composed with $\mathbf{R}$ and $\mathbf{T}$, as well as the intrinsic matrix $\mathbf{I}$ composed with $F$ and $O_x, O_y$. Given Gaussian Ellipsoids, VoGE renderer synthesizes an image $\mathbf{O}$.

### 3.3 RENDER GAUSSIAN ELLIPSOIDS

Figure 4 shows the rendering process for VoGE. VoGE takes inputs of a perspective camera and Gaussian ellipsoids to render images, while computing gradient towards both camera and Gaussian ellipsoids (shows in Figure 2 and 3).

**Viewing transformation** utilizes the extrinsic configuration $\mathbf{E}$ of the camera to transfer the Gaussian ellipsoids from the object coordinate to the camera coordinate. Let $\mathbf{M}_k^o$ denote centers of ellipsoids in the object coordinate. Following the standard approach, we compute the centers in the camera coordinate:

$$\mathbf{M}_k = \mathbf{R} \cdot \mathbf{M}_k^o + \mathbf{T} \tag{3}$$

where $\mathbf{R}$ and $\mathbf{T}$ are the rotation and translation matrix included in $\mathbf{E}$. Since we consider 3D Gaussian Kernels are ellipsoidal, observations of the variance matrices are also changed upon camera rotations:

$$\mathbf{\Sigma}_k^{-1} = \mathbf{R}^T \cdot (\mathbf{\Sigma}_k^o)^{-1} \cdot \mathbf{R} \tag{4}$$

**Perspective rays** indicate the viewing direction in the camera coordinate. For each pixel, we compute the viewing ray under the assumption that the camera is fully perspective:

$$\mathbf{r}(t) = \mathbf{D} * t = \begin{bmatrix} \frac{i-O_y}{F} & \frac{j-O_x}{F} & 1 \end{bmatrix}^T * t \tag{5}$$

where $p = (i, j)$ is the pixel location on the image, $O_x, O_y$ is the principal point of the camera, $F$ is the focal length, $\mathbf{D}$ is the ray direction vector.

**Ray tracing** observes the volume densities of each ellipsoid along the ray $\mathbf{r}$ respectively. Note the observation of each ellipsoid is a 1D Gaussian function along the viewing ray (for detailed mathematics, refer to Appendix A.1):

$$\rho_m(\mathbf{r}(s)) = \exp(q_m - \frac{(s - l_m)^2}{2 \cdot \sigma_m^2}) \tag{6}$$

where $l_m = \frac{\mathbf{M}_m^T \cdot \mathbf{\Sigma}_m^{-1} \cdot \mathbf{D} + \mathbf{D}^T \cdot \mathbf{\Sigma}_m^{-1} \cdot \mathbf{M}_m}{2 \cdot \mathbf{D}^T \cdot \mathbf{\Sigma}_m^{-1} \cdot \mathbf{D}}$ is the length along the ray that gives peak activation for $m$-th kernel. $q_m = -\frac{1}{2}\mathbf{V}_m^T \cdot \mathbf{\Sigma}_m^{-1} \cdot \mathbf{V}_m, where\ \mathbf{V}_m = \mathbf{M}_m - l_m \cdot \mathbf{D}$ computes peak density of $m$-th kernel alone the ray. The 1D variance is computed via $\frac{1}{\sigma_m^2} = \mathbf{D}^T \cdot \mathbf{\Sigma}_m^{-1} \cdot \mathbf{D}$. Thus, when tracing along each ray, we only need to record $l_m$, $q_m$ and $\sigma_m$ for each ellipsoid respectively.

**Blending via Volume Densities** computes the observation along the ray $\mathbf{r}$. As Figure 1 shows, different from other generic renderers, which only consider the viewing distance for blending, VoGE blends all observations based on the integral of volume densities along the ray. However, computing the integral using brute force is so computationally inefficient that even infeasible for concurrent computation power. To resolve this, we propose an approximate closed-form solution, which conducts the computation in both an accurate and effective way. We use the Error Function erf to compute the integral of Gaussian, since it can be computed via a numerical approach directly. Specifically, with Equation 2 and Equation 5, we can calculate the transmittance $T(t)$ as (for proof about this approximation, refer to Appendix A.2):

$$T(t) = \exp(-\tau \int_{-\infty}^{t} \rho(\mathbf{r}(s))ds) = \exp(-\tau \sum_{m=1}^{K} e^{q_m} \frac{\text{erf}((t - l_m)/\sigma_m) + 1}{2}) \tag{7}$$

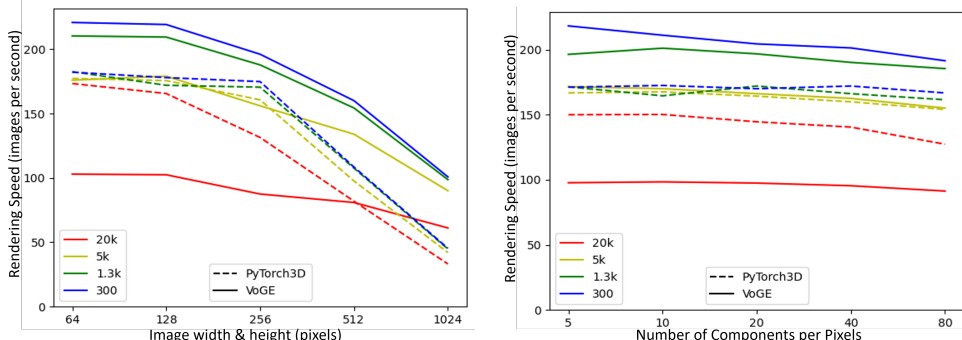

Figure 5: Comparison for rendering speeds of VoGE and PyTorch3D, reported in images per second (higher better). We evaluate the rendering speed using cuboids with different number of primitives (vertices, ellipsoids), which illustrated using different colors, also different image sizes and number of primitives per pixel.

Now, to compute closed-form solution of the outer integral in Equation 1, we use the $T(t), t = l_k$ at the peak of $\rho(\mathbf{r}(t))$ alone the rays. Here we provide the closed-form solution for $C(\mathbf{r})$:

$$C(\mathbf{r}) = \int_{-\infty}^{\infty} T(t)\rho(\mathbf{r}(t))\mathbf{c}(\mathbf{r}(t))dt = \sum_{k=1}^{K} T(l_k)e^{q_k}\mathbf{c}_k \tag{8}$$

Note based on the assumption that distances from camera to ellipsoids are significantly larger than ellipsoid sizes, thus it is equivalent to set $t_n = -\infty$ and $t_f = \infty$.

**Coarse-to-fine rendering.** In order to improve the rendering efficiency, we implement VoGE rendering with a coarse-to-fine strategy. Specifically, VoGE renderer has an optional coarse rasterizing stage that, for each ray, selects only around 10% of all ellipsoids (details in Appendix A.3). Besides, the ray tracing volume densities also works in a coarse-to-fine manner. VoGE blends $K^{'}$ nearest ellipsoids among all traced kernels that gives $e^{q_k} > thr = 0.01$. Using CUDA from NVIDIA et al. (2022), we implement VoGE with both forward and backward function. The CUDA-VoGE is packed as an easy-to-use "autogradable" PyTorch API.

### 3.4 VoGE in Neural Networks

VoGE can be easily embedded into neural networks by serving as neural sampler and renderer. As a sampler, VoGE extracts attributes $\alpha_k$ (*e.g.*, deep neural features, textures) from images or feature maps into kernel-correspond attributes, which is conducted via reconstructing their spatial distribution in the screen coordinates. When serving as a renderer, VoGE converts kernel-correspond attributes into images or feature maps. Since both sampling and rendering give the same spatial distribution of feature/texture, it is possible for VoGE to conduct geometry-based image-to-image transformation.

Here we discuss how VoGE samples deep neural features. Let $\boldsymbol{\Phi}$ denotes observed features, where $\phi_p$ is the value at location $p$. Let $\mathbf{A} = \bigcup_{k=1}^{K}\{\alpha_k\}$ denotes the per kernel attribute, which we want to discover during sampling. With a given object geometry $\boldsymbol{\Gamma} = \bigcup_{k=1}^{K}\{\mathbf{M}_k, \boldsymbol{\Sigma}_k\}$ and viewing rays $\mathbf{r}(p)$. The the observation formulated with conditional probability regarding $\alpha_k$:

$$\phi^{'}(p) = \sum_{k=1}^{K} \mathcal{P}(\alpha_k|\boldsymbol{\Gamma}, \mathbf{r}(p), k)\alpha_k \tag{9}$$

Since $\boldsymbol{\Phi}$ is a discrete observation of a continuous distribution $\phi(p)$ on the screen, the synthesis can only be evaluated at discrete positions, *i.e.* the pixel centers. As the goal is to make $\boldsymbol{\Phi}^{'}$ similar as $\boldsymbol{\Phi}$ on all observable locations, we resolve via an inverse reconstruction:

$$\alpha_k = \sum_{p=1}^{P} \mathcal{P}(\phi(p)|\boldsymbol{\Gamma}, \mathbf{r}(p), p)\phi(p) = \frac{\sum_{p=1}^{P}\mathbf{W}_{p,k} * \phi_p}{\sum_{p=1}^{P}\mathbf{W}_{p,k}} \tag{10}$$

where $\mathbf{W}_{p,k} = T(l_k)e^{q_k}$ is the kernel-to-pixel weight as described in 3.1.

Table 2: Pose estimation results on the PASCAL3D+ and the Occluded PASCAL3D+ dataset. Occlusion level L0 is the original images from PASCAL3D+, while Occlusion Level L1 to L3 are the occluded PASCAL3D+ images with increasing occlusion ratios. NeMo is an object pose estimation pipeline via neural feature level render-and-compare. We compare the object pose estimation performance using different renderers, *i.e.* VoGE, Soft Rasterizer, DSS, PyTorch3D (which is used in NeMo originally).

| Evaluation Metric | $ACC_{\frac{\pi}{6}} \uparrow$ | | | | $ACC_{\frac{\pi}{18}} \uparrow$ | | | | $MedErr \downarrow$ | | | |
|---|---|---|---|---|---|---|---|---|---|---|---|---|
| Occlusion Level | L0 | L1 | L2 | L3 | L0 | L1 | L2 | L3 | L0 | L1 | L2 | L3 |
| Res50-General | 88.1 | 70.4 | 52.8 | 37.8 | 44.6 | 25.3 | 14.5 | 6.7 | 11.7 | 17.9 | 30.4 | 46.4 |
| Res50-Specific | 87.6 | 73.2 | 58.4 | 43.1 | 43.9 | 28.1 | 18.6 | 9.9 | 11.8 | 17.3 | 26.1 | 44.0 |
| StarMap | 89.4 | 71.1 | 47.2 | 22.9 | 59.5 | 34.4 | 13.9 | 3.7 | 9.0 | 17.6 | 34.1 | 63.0 |
| NeMo+SoftRas | 85.3 | 75.2 | 63.0 | 44.3 | 59.7 | 46.7 | 32.1 | 16.8 | 9.1 | 14.8 | 24.0 | 39.3 |
| NeMo+DSS | 81.1 | 71.9 | 56.8 | 38.7 | 33.5 | 30.4 | 23.0 | 14.1 | 16.1 | 19.8 | 25.8 | 40.4 |
| NeMo+PyTorch3D | 86.1 | 76.0 | 63.9 | 46.8 | 61.0 | 46.3 | 32.0 | 17.1 | 8.8 | 13.6 | 20.9 | 36.5 |
| NeMo+VoGE(Ours) | **90.1** | **83.1** | **72.5** | **56.0** | **69.2** | **56.1** | **41.5** | **24.8** | **6.9** | **9.9** | **15.0** | **26.3** |

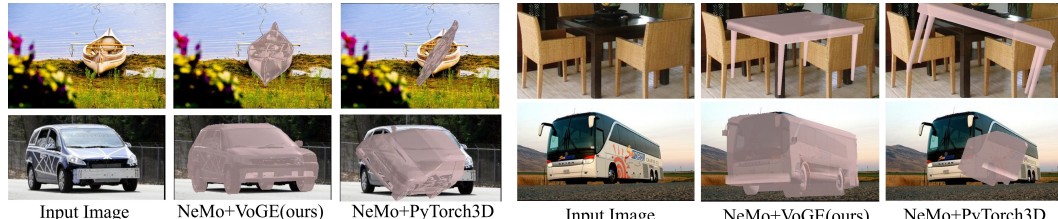

| Input Image | NeMo+VoGE(ours) | NeMo+PyTorch3D | | Input Image | NeMo+VoGE(ours) | NeMo+PyTorch3D |

Figure 6: Qualitative object pose estimation results on PASCAL3D+ dataset. We visualize the predicted object poses from NeMo+VoGE and standard NeMo. Specifically, we use a standard mesh renderer to render the original CAD model under the predicted pose and superimpose onto the input image.

# 4 EXPERIMENT

We explore several applications of VoGE. In section 4.1, we study the object pose estimation using VoGE in a feature level render-and-compare pose estimator. In section 4.2, we explore texture extraction ability of VoGE. In section 4.4, we demonstrate VoGE can optimize the shape representation via multi-viewed images. Visualizations of VoGE rendering are included in Appendix B.

**Rendering Speed.** As Figure 5 shows, CUDA-VoGE provides a competitive rendering speed compare to state-of-the-art differentiable generic renderer when rendering a single cuboidal object.

## 4.1 OBJECT POSE ESTIMATION IN WILD

We evaluate the ability of VoGE when serving as a feature sampler and renderer in an object pose estimation pipeline, NeMo Wang et al. (2020a), an in-wild category-level object 3D pose estimator that conducts render-and-compare on neural feature level. NeMo utilizes PyTorch3D Ravi et al. (2020) as the feature sampler and renderer, which converts the feature maps to vertex corresponded feature vectors and conducts the inverse process. In our NeMo+VoGE experiment, we use VoGE to replace the PyTorch3D sampler and renderer via the approach described in Section 3.4.

**Dataset.** Following NeMo, we evaluate pose estimation performance on the PASCAL3D+ dataset Xiang et al. (2014), the Occluded PASCAL3D+ dataset Wang et al. (2020b) and the ObjectNet3D dataset Xiang et al. (2016). The PASCAL3D+ dataset contains objects in 12 man-made categories with 11045 training images and 10812 testing images. The Occluded PASCAL3D+ contains the occluded version of same images, which is obtained via superimposing occluder cropped from MS-COCO dataset Lin et al. (2014). The dataset includes three levels of occlusion with increasing occlusion rates. In the experiment on ObjectNet3D, we follow NeMo to test on 18 categories.

**Evaluation Metric.** We measure the pose estimation performance via accuracy of rotation error under given thresholds and median of per image rotation errors. The rotation error is defined as the difference between the predicted rotation matrix and the ground truth rotation matrix:

$$\Delta(R_{pred}, R_{gt}) = \frac{\left\| \log m\left( R_{pred}^T R_{gt} \right) \right\|_F}{\sqrt{2}}$$

Table 3: Pose estimation results on the ObjectNet3D dataset. Evaluated via pose estimation accuracy for error under $\frac{\pi}{6}$ (higher better).

| $ACC_{\frac{\pi}{6}} \uparrow$ | bed | shelf | calculator | cellphone | computer | cabinet | guitar | iron | knife |
|---|---|---|---|---|---|---|---|---|---|
| StarMap | 40.0 | 72.9 | 21.1 | 41.9 | 62.1 | 79.9 | 38.7 | 2.0 | 6.1 |
| NeMo+PyTorch3D | 56.1 | 53.7 | 57.1 | 28.2 | **78.8** | **83.6** | 38.8 | 32.3 | 9.8 |
| NeMo+VoGE(Ours) | **76.8** | **83.2** | **77.8** | **50.7** | **78.8** | **83.6** | **54.6** | **45.4** | **12.1** |
| $ACC_{\frac{\pi}{6}} \uparrow$ | oven | pen | pot | rifle | slipper | stove | toilet | tub | wheelchair |
| StarMap | 86.9 | 12.4 | 45.1 | 3.0 | 13.3 | 79.7 | 35.6 | 46.4 | 17.7 |
| NeMo+PyTorch3D | 90.3 | 3.7 | 66.7 | 13.7 | 6.1 | 85.2 | 74.5 | 61.6 | **71.7** |
| NeMo+VoGE(Ours) | **94.9** | **13.5** | **77.8** | **30.8** | **22.2** | **89.8** | **81.9** | **68.9** | 68.4 |

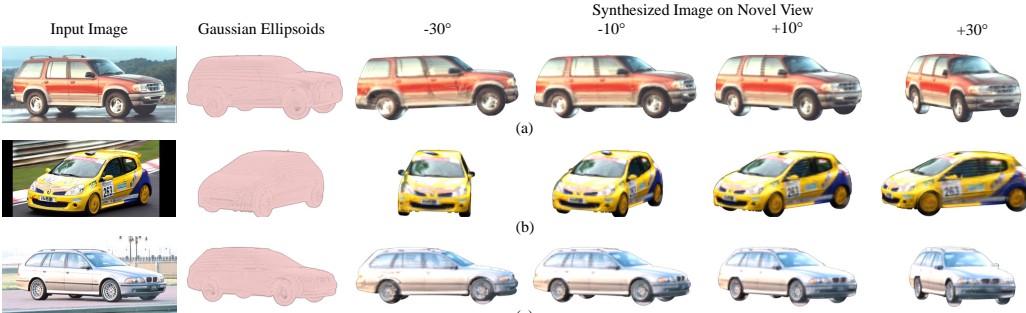

Figure 7: Sampling texture and rerendering on novel view. The inputs include a single RGB image and the Gaussian Ellipsoids with corresponded pose. Note the result is produced **without** any training or symmetrical information.

**Baselines.** We compare our VoGE for object pose estimation with other state-of-the-art differentiable renderers, *i.e.* Soft Rasterizer, DSS, and PyTorch3D. For comparison, we use the same training and inference pipeline, and same hyper-parameters for all 4 experiments. Our baselines also includes Res50-General/Specific which converts object pose estimation into a bin classification problem, and StarMap Zhou et al. (2018) which first detect keypoints and conduct pose estimation via the PnP method.

**Experiment Details.** Following the experiment setup in NeMo, we train the feature extractor 800 epochs with a progressive learning rate. During inference, for each image, we sample 144 starting poses and optimizer 300 steps via an ADAM optimizer. We convert the meshes provided by NeMo using the method described Section 3.2.

**Results.** Figure 6 and Table 2 show the qualitative and quantitative results of object pose estimation on PASCAL3D+ and the Occluded PASCAL3D+ dataset. Results in Table 2 demonstrate significant performance improvement using VoGE compared to Soft RasterizerLiu et al. (2019), DSSYifan et al. (2019) and PyTorch3DRavi et al. (2020). Moreover, both qualitative and quantitative results show our method a significant robustness under partial occlusion and out distributed cases. Also, Figure 6 demonstrates our approach can generalize to those out distributed cases, *e.g.*, a car without front bumper, while infeasible for baseline renderers. Table 3 shows the results on ObjectNet3D, which demonstrates a significant performance gain compared to the baseline approaches. The ablation study is included in Appendix C.1.

### 4.2 TEXTURE EXTRACTION AND RERENDERING

As Figure 7 shows, we conduct the texture extraction on real images and rerender the extracted textures under novel viewpoints. The qualitative results is produced on PASCAL3D+ dataset. The experiment is conducted on each image independently that there is no training included. Specifically, for each image, we have only three inputs, *i.e.* the image, the camera configuration, the Gaussian ellipsoids converted from the CAD models provided by the dataset. Using the method proposed in 3.4, we extract the RGB value for each kernel on the Gaussian ellipsoids using the given groundtruth camera configuration. Then we rerender Gaussian ellipsoids with the extracted texture under a novel view, that we increase or decrease the azimuth of the viewpoint (horizontal rotation). The qualitative

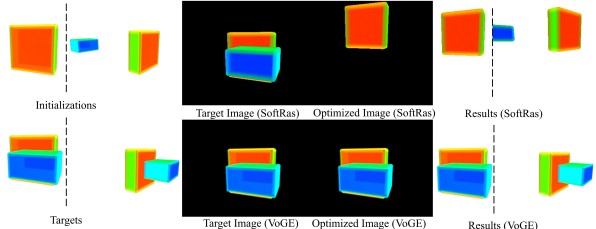
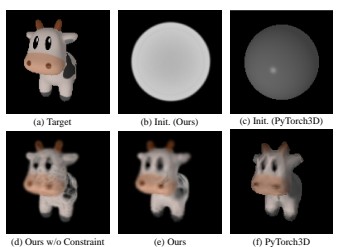

Figure 8: Reasoning multi-object occlusions for single view optimization of object location. Left: initialization and target locations. Middle: target images, note the target image generate via each rendering method. Right: results. Videos: VoGE, SoftRas.

Figure 9: Shape fitting results with 20 multi-viewed images following the PyTorch3D Ravi et al. (2020) official tutorial.

results demonstrate a satisfying texture extraction ability of VoGE, even with only a single image. Also, the details (*e.g.*, numbers on the second car) are retained in high quality under the novel views.

### 4.3    OCCLUSION REASONING OF MULTIPLE OBJECTS

Figure 8 shows differentiating the occlusion reasoning process between two objects. Specifically, a target image, and the colored cuboid models and initialization locations, are given to the method. Then we render and optimize the 3D locations of both the cuboids. In this experiment, we find both SoftRas and VoGE can successfully optimize the locations when the occludee (blue cuboid) is near the occluder (red cuboid), which is 1.5 scales behind the occluder as the thickness of the occluder is 0.6 scales. However, when the the occludee is far behind the occluder (5 scales), SoftRas fails to produce correct gradient to optimize the locations, whereas VoGE can still successfully optimize the locations. We think such advantage benefits from the better volume density blending compared to the distance based blender used in SoftRas.

### 4.4    SHAPE FITTING VIA INVERSE RENDERING

Figure 9 shows the qualitative results of multi-viewed shape fitting. In this experiment, we follows the setup in *fit a mesh with texture via rendering* from PyTorch3D official tutorial Ravi et al. (2022a). First, a standard graphic renderer is used to render the cow CAD model in 20 different viewpoints under a fixed light condition, which are used as the optimization targets. For both baseline and ours, we give a sphere object geometry with 2562 vertices and optimize toward target images using the same configuration, *e.g.*, iterations, learning rate, optimizer, loss function. During the shape optimization process, we compute MSE loss on both silhouettes and RGB values between the synthesized images and the targets. The vertices locations and colors are gradiently updated with an ADAM optimizer Kingma & Ba (2014). We conduct the optimization for 2000 iterations, while in each iteration, we randomly select 5 out of 20 images to conduct the optimization. In Figure 9 (e) and (f), we use the normal consistency, edge and Laplacian loss Nealen et al. (2006) to constrain the object geometry, while in (d) no additional loss is used. From the results, we can see that VoGE has a competitive ability regarding shape fit via deformation. Specifically, VoGE gives better color prediction and a smoother object boundary. Also, we observe the current geometry constrain losses do not significantly contribute to our final prediction. We argue those losses are designed for surface triangular meshes, that not suitable for Gaussian ellipsoids. The design of geometry constraints that are suitable for Gaussian ellipsoids is an interesting topic but beyond scope of this paper.

## 5    CONCLUSION

In this work, we propose VoGE, a differentiable volume renderer using Gaussian Ellipsoids. Experiments on in-wild object pose estimation and neural view matching show VoGE an extraordinary ability when applied on neural features compare to the concurrent famous differential generic renderers. Texture extraction and rerendering experiment shows VoGE the ability on feature and texture sampling, which potentially benefits downstream tasks. Overall, VoGE demonstrates better differentiability, which benefits vision tasks, while retains competitive rendering speed.

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

# A    ADDITIONAL DETAILS OF VOGE RENDERER

In this section we provide more detailed discussion for the math of ray tracing volume densities in VoGE (section A.1 and A.2), coarse-to-fine rendering strategy (section A.3), and the converters (section A.4).

## A.1    RAY TRACING

In this section, we provide the detailed deduction process for Equations 6 in the main text. First, let's recall the formula of Ray tracing volume densities Kajiya & Von Herzen (1984):

$$C(\mathbf{r}) = \int_{t_n}^{t_f} T(t)\rho(\mathbf{r}(t))\mathbf{c}(\mathbf{r}(t))dt,$$
$$\text{where } T(t) = \exp\left(-\tau \int_{t_n}^{t} \rho(\mathbf{r}(s))ds\right) \tag{11}$$

where $T(t)$ is the occupancy function alone viewing ray $\mathbf{r}(t)$, as we describe in Equation 5 in main text:

$$\mathbf{r}(t) = \mathbf{D} * t \tag{12}$$

where $\mathbf{D}$ is the normalized direction vector of the viewing ray.

Also, as we describe in Section 3.2, we reconstruct the volume density function $\rho(\mathbf{r}(t))$ via the sum of a set of ellipsoidal Gaussians:

$$\rho(\mathbf{X}) = \sum_{k=1}^{K} \frac{1}{\sqrt{2\pi \cdot ||\mathbf{\Sigma}_k||_2}} e^{-\frac{1}{2}(\mathbf{X}-\mathbf{M}_k)^T \cdot \mathbf{\Sigma}_k^{-1} \cdot (\mathbf{X}-\mathbf{M}_k)} \tag{13}$$

where $K$ is the total number of Gaussian kernels, $\mathbf{X} = (x, y, z)$ is an arbitrary location in the 3D volume. $\mathbf{M}_k$ is the center of $k$-th ellipsoidal Gaussians kernel:

$$\mathbf{M}_k = (\mu_{k,x}, \mu_{k,y}, \mu_{k,z}) \tag{14}$$

whereas the $\mathbf{\Sigma}_k$ is the spatial variance matrix:

$$\mathbf{\Sigma}_k = \begin{bmatrix} \sigma_{k,xx} & \sigma_{k,xy} & \sigma_{k,xz} \\ \sigma_{k,yx} & \sigma_{k,yy} & \sigma_{k,yz} \\ \sigma_{k,zx} & \sigma_{k,zy} & \sigma_{k,zz} \end{bmatrix} \tag{15}$$

Note that $\mathbf{\Sigma}_k$ is a symmetry matrix, e.g., covariance $\sigma_{k,xy} = \sigma_{k,yx}$.

**Occupancy Function.** Based on Equation 13 and 11, $T(t)$ can be computed via:

$$T(t) = \exp\left(-\tau \int_{t_n}^{t} \rho(\mathbf{r}(s))ds\right)$$
$$= \exp(-\tau \int_{t_n}^{t} \sum_{k=1}^{K} \frac{1}{\sqrt{2\pi \cdot ||\mathbf{\Sigma}_k||_2}} e^{-\frac{1}{2}(s\mathbf{D}-\mathbf{M}_k)^T \cdot \mathbf{\Sigma}_k^{-1} \cdot (s\mathbf{D}-\mathbf{M}_k)}ds) \tag{16}$$

Now, let $\mathbf{M}_k = l_k\mathbf{D} + \mathbf{V}_k$, where $l_k$ is a length along the viewing ray, $\mathbf{V}_k = \mathbf{M}_k - l_k\mathbf{D}$ is the vector from location $l_k\mathbf{D}$ on the ray to the vertex $\mathbf{M}_k$ (we will discuss a solution for $\mathbf{V}_k$ and $l_k$ later). Equation 16 can be simplified as:

$$T(t) = \exp(-\tau \int_{t_n}^{t} \sum_{k=1}^{K} \frac{1}{\sqrt{2\pi \cdot ||\mathbf{\Sigma}_k||_2}} e^{-\frac{1}{2}(s-l_k)^2\mathbf{D}^T \cdot \mathbf{\Sigma}_k^{-1} \cdot \mathbf{D}}$$
$$e^{-\frac{1}{2}(s-l_k)(\mathbf{V}_k^T \cdot \mathbf{\Sigma}_k^{-1} \cdot \mathbf{D} + \mathbf{D}^T \cdot \mathbf{\Sigma}_k^{-1} \cdot \mathbf{V}_k)} e^{-\frac{1}{2}\mathbf{V}_k^T \cdot \mathbf{\Sigma}_k^{-1} \cdot \mathbf{V}_k}ds) \tag{17}$$

In order to further simplify $T(t)$, we take $\mathbf{V}_k$ that makes:

$$\mathbf{V}_k^T \cdot \mathbf{\Sigma}_k^{-1} \cdot \mathbf{D} + \mathbf{D}^T \cdot \mathbf{\Sigma}_k^{-1} \cdot \mathbf{V}_k = 0 \tag{18}$$

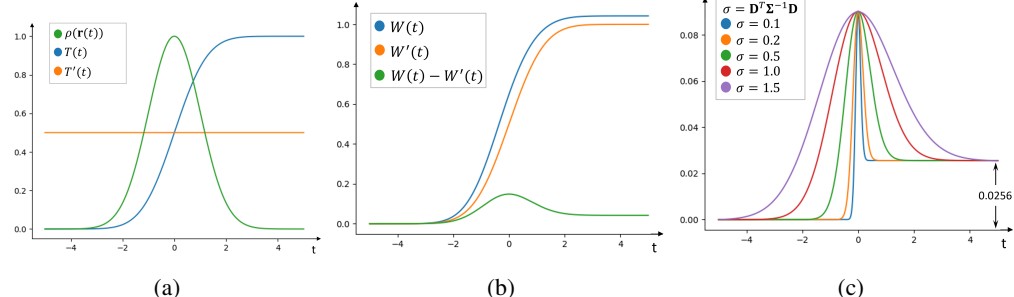

Figure 10: Approximate computation of integral along the viewing ray for a single kernel. (a) $T(t)$ is the real occupancy function along the ray, $T'(t)$ means we use the occupancy at $l_k$ of $\rho(\mathbf{r}(t))$, since $\rho(\mathbf{r}(t))$ are mainly concentrate near $l_k$. (b) shows $W(t) = \int_{-\infty}^t T(s) \cdot \rho(\mathbf{r}(s))ds$, $W'(t) = \int_{-\infty}^t T'(s) \cdot \rho(\mathbf{r}(s))ds$ and the difference $W(t) - W'(t)$. Since we use the infinite integral with $t$, only error at end of $t$ axis need to be consider. (c) shows the accumulative $W(t) - W'(t)$ using different $\sigma = \mathbf{D}^T \cdot \mathbf{\Sigma}^{-1} \cdot \mathbf{D}$. Interestingly, the final error gives a fix value which is independent from $\sigma$. Note that the final error is $0.0256$ which can be ignored when compared to the integral result $W = 1$.

which can be solve using $\mathbf{V}_k = \mathbf{M}_k - l_k\mathbf{D}$:

$$
\begin{aligned}
(\mathbf{M}_k - l_k\mathbf{D})^T \cdot \mathbf{\Sigma}_k^{-1} \cdot \mathbf{D} + \mathbf{D}^T \cdot \mathbf{\Sigma}_k^{-1} \cdot (\mathbf{M}_k - l_k\mathbf{D}) &= 0 \\
\mathbf{M}_k^T \cdot \mathbf{\Sigma}_k^{-1} \cdot \mathbf{D} + \mathbf{D}^T \cdot \mathbf{\Sigma}_k^{-1} \cdot \mathbf{M}_k - 2l_k\mathbf{D}^T \cdot \mathbf{\Sigma}_k^{-1} \cdot \mathbf{D} &= 0 \\
l_k = \frac{\mathbf{M}_k^T \cdot \mathbf{\Sigma}_k^{-1} \cdot \mathbf{D} + \mathbf{D}^T \cdot \mathbf{\Sigma}_k^{-1} \cdot \mathbf{M}_k}{2 \cdot \mathbf{D}^T \cdot \mathbf{\Sigma}_k^{-1} \cdot \mathbf{D}}
\end{aligned}
\tag{19}
$$

Note that $l_k$ is also the length that gives the maximum density $\rho_k(\mathbf{r}(t))$ along the ray for $k$-th kernel. To proof this, we compute:

$$
\begin{aligned}
\frac{\partial}{\partial t}\rho_k(\mathbf{r}(t)) &= \frac{\partial}{\partial t} \frac{1}{\sqrt{2\pi \cdot ||\mathbf{\Sigma}_k||_2}} e^{-\frac{1}{2}(t\mathbf{D}-\mathbf{M}_k)^T \cdot \mathbf{\Sigma}_k^{-1} \cdot (t\mathbf{D}-\mathbf{M}_k)} \\
&= \frac{1}{4\sqrt{2\pi ||\mathbf{\Sigma}_k||_2}} (t\mathbf{D} - \mathbf{M}_k)^T \mathbf{\Sigma}_k^{-1}(t\mathbf{D} - \mathbf{M}_k) \\
&\quad \cdot (\mathbf{M}_k^T\mathbf{\Sigma}_k^{-1}\mathbf{D} + \mathbf{D}^T\mathbf{\Sigma}_k^{-1}\mathbf{M}_k - 2t\mathbf{D}^T\mathbf{\Sigma}_k^{-1}\mathbf{D}) \\
&\quad \cdot e^{-\frac{1}{2}(t\mathbf{D}-\mathbf{M}_k)^T \cdot \mathbf{\Sigma}_k^{-1} \cdot (t\mathbf{D}-\mathbf{M}_k)}
\end{aligned}
\tag{20}
$$

Obviously, the solve for $\frac{\partial}{\partial t}\rho_k(\mathbf{r}(t)) = 0$ is:

$$
t = \frac{\mathbf{M}_k^T \cdot \mathbf{\Sigma}_k^{-1} \cdot \mathbf{D} + \mathbf{D}^T \cdot \mathbf{\Sigma}_k^{-1} \cdot \mathbf{M}_k}{2 \cdot \mathbf{D}^T \cdot \mathbf{\Sigma}_k^{-1} \cdot \mathbf{D}} = l_k
\tag{21}
$$

Now the density function of the $k$-th ellipsoid along the viewing ray $\mathbf{r}(s)$ gives an 1D Gaussian function:

$$
\begin{aligned}
\rho_k(\mathbf{r}(s)) &= \frac{e^{-\frac{1}{2}\mathbf{V}_k^T \cdot \mathbf{\Sigma}_k^{-1} \cdot \mathbf{V}_k}}{\sqrt{2\pi \cdot ||\mathbf{\Sigma}_k||_2}} e^{-\frac{1}{2}(s-l_k)^2 \mathbf{D}^T \cdot \mathbf{\Sigma}_k^{-1} \cdot \mathbf{D}} \\
&= \frac{1}{\sqrt{2\pi \cdot ||\mathbf{\Sigma}_k||_2}} \cdot \exp(q_k - \frac{(s-l_k)^2}{2 \cdot \sigma_k^2})
\end{aligned}
\tag{22}
$$

where $q_k = -\frac{1}{2}\mathbf{V}_k^T \cdot \mathbf{\Sigma}_k^{-1} \cdot \mathbf{V}_k$, $\frac{1}{\sigma_k^2} = \mathbf{D}^T \cdot \mathbf{\Sigma}_k^{-1} \cdot \mathbf{D}$. Thus, when tracing along each ray, we only need to record $l_k$, $q_k$ and $\sigma_k$ for each ellipsoid respectively.

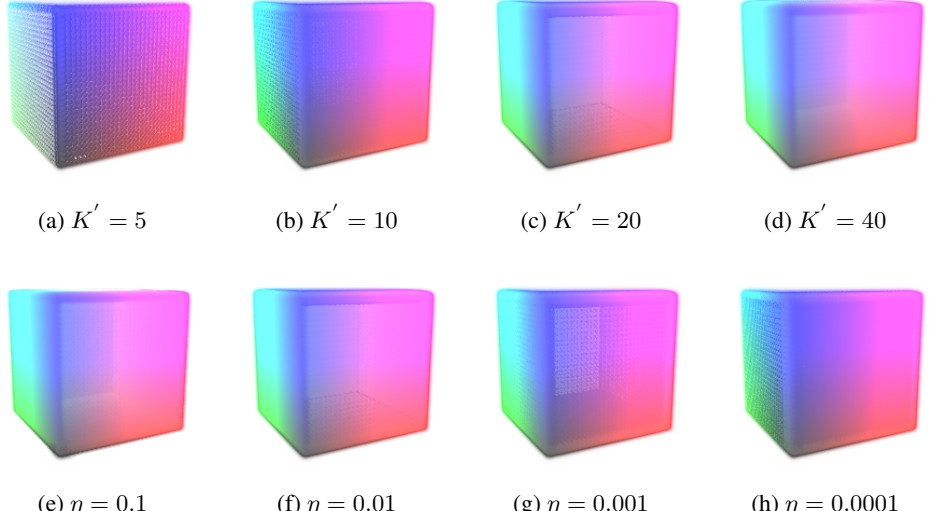

(a) $K^{'} = 5$      (b) $K^{'} = 10$      (c) $K^{'} = 20$      (d) $K^{'} = 40$

(e) $\eta = 0.1$      (f) $\eta = 0.01$      (g) $\eta = 0.001$      (h) $\eta = 0.0001$

Figure 11: Rendering cuboid using different $K^{'}$ and $\eta$. (a) to (d) shows the rendering result using different $K^{'}$, the threshold is fix as $\eta = 0.01$. (e) to (h) shows the rendered cuboid with different $\eta$, while fixing $K^{'} = 20$.

### A.2 BLENDING VIA VOLUME DENSITY

Since $q_k$ is independent from $t$, the Equation 16 can be further simplified:

$$
\begin{aligned}
T(t) &= \exp(-\sum_{k=1}^{K} e^{q_k} \int_{-\infty}^{t} \frac{1}{\sqrt{2\pi \cdot ||\mathbf{\Sigma}_k||_2}} e^{-(s-l_k)^2/\sigma_k^2} ds) \\
&= \exp(-\sum_{k=1}^{K} e^{q_k} \frac{\mathrm{erf}((t-l_k)/\sigma_k)+1}{2})
\end{aligned}
\tag{23}
$$

where $\mathrm{erf}$ is the error function, that concurrent computation platforms, *e.g.*, PyTorch, Scipy, have already implemented.

**Scattering Equation.** Now we compute the final color observation $C(\mathbf{r})$. As we describe in Section 3.2, we assume each kernel has a homogeneous $\mathbf{C}_k$. Thus, here we compute:

$$
\begin{aligned}
\mathbf{W}_k(t) &= \int_{t_n}^{t_f} T(t)\rho(\mathbf{r}(t)) dt \\
&= \int_{-\infty}^{\infty} T(t) \sum_{k=1}^{K} \frac{1}{\sqrt{2\pi \cdot ||\mathbf{\Sigma}_k||_2}} e^{-\frac{1}{2}(\mathbf{X}-\mathbf{M}_k)^T \cdot \mathbf{\Sigma}_k^{-1} \cdot (\mathbf{X}-\mathbf{M}_k)} dt
\end{aligned}
\tag{24}
$$

where $\mathbf{X} = t\mathbf{D}$. Similar to previous simplifications, we use $q_k$ and $l_k$ to replace $\mathbf{M}_k$ in Equation 24:

$$
\mathbf{W}_k(t) = \sum_{k=1}^{K} e^{q_k} \int_{-\infty}^{\infty} T(t) \frac{1}{\sqrt{2\pi \cdot ||\mathbf{\Sigma}_k||_2}} e^{-(t-l_k)^2/\sigma_k^2} dt
\tag{25}
$$

Due to the error function is already a complex function, it is infeasible to compute the integral of $T(t)$. We propose an approximate solution that we use $T(l_k)$ to replace $T(t)$ inside the integral. Now the final closed-form solution for $\mathbf{W}_k(t)$ is computed by:

$$
\begin{aligned}
\mathbf{W}_k(t) &= \sum_{k=1}^{K} T(l_k) e^{q_k} \int_{-\infty}^{\infty} \frac{1}{\sqrt{2\pi \cdot ||\mathbf{\Sigma}_k||_2}} e^{-(t-l_k)^2/\sigma_k^2} dt \\
&= \sum_{k=1}^{K} T(l_k) e^{q_k}
\end{aligned}
\tag{26}
$$

Because of the complexity when computing integral of the erf function, here we prove that in practice such approximate gives high enough accuracy. To simplify the problem, we study the case that the volume only contains a single Gaussian ellipsoid kernel. We further suggest that in the multi-kernel cases, the errors between different kernels introduced by the approximation will be lower. Because $m$-th kernel has a low $\rho_m(\mathbf{r}(t))$ at $l_k$, which makes the corresponded $T(t)$ more flatten, thus the approximation fits better. As Figure 10 (a) shows, we plot density function along the ray. Specifically, we sample 10k points on the ray, and for each point, we plot its density, the real occupancy, and the approximate occupancy. Figure 10 (b) shows the real weight $W$ which is computed via the cumulative sum along the ray, and the approximate weight $W'$ which is computed via our proposed approximate closed-from solution. We also show the difference between $W'$ and $W$ with the green line, which is significantly smaller compare to $W$. Interestingly, as Figure 10 (c) shows, we find the error $W(t) - W'(t)$ is independent from $\mathbf{\Sigma}^{-1}$ and $\mathbf{D}$, that always converge to a same value: 0.0256. Though we cannot give a mathematical explanation regarding this phenomenon, we argue the result is already enough to draw the conclusion that such approximation gives satisfying accuracy.

### A.3 COARSE-TO-FINE RENDERING WITH KERNEL SELECTION

As we discussed in Section 3.3 in the main text, in order to efficiently render Gaussian ellipsoids, we design the coarse-to-fine rendering strategy. Specifically, we gradually reduce the number of ellipsoids that interact with viewing rays. Following PyTorch3D, we develop a optional coarse rasterization stage, which select 10% of all ellipsoids and feed them into the ray tracing stage. Specifically, we project the center of each ellipsoid onto the screen coordinate via standard object-to-camera transformation, then for each ellipsoids, we compute the height $b_h$ and width $b_w$ of a maximum bounding box of the ellipsoids in 2D screen coordinate. The height and width are computed via:

$$[b_h \quad b_w \quad .] = \frac{\log(-\eta)}{d_z} \cdot \mathbf{\Omega} \cdot \mathbf{\Sigma}^{-1} \cdot \mathbf{\Omega} \tag{27}$$

where $d_z$ is the distance from camera to the center of ellipsoid, $\eta$ is the threshold for maximum volume density, $\mathbf{\Omega}$ is the projection matrix from camera coordinate to screen:

$$\mathbf{\Omega} = \begin{bmatrix} \frac{2 \cdot F}{h} & 0 & 0 \\ 0 & \frac{2 \cdot F}{w} & 0 \\ 0 & 0 & 1 \end{bmatrix} \tag{28}$$

Then we rasterize the bounding boxes to produce a pixel-to-kernels assignment in a low resolution (8 times smaller compared to the image size), which indicates the set of ellipsoid kernels for each pixel to trace.

Similarly, the ray tracing stage is also select only part of all Gaussian ellipsoids to feed into the blending stage. When conducting ray tracing, we only trace $K'$ nearest kernels that has non-trivial contributions regarding its final weight $\mathbf{W}_k$. Specifically, we first record all ellipsoids that gives a maximum density $e^{q_k} > \eta$. For all the recorded kernels, we sort them via the length to the 1D Gaussian center $l_k$ and select $K'$ nearest ellipsoids. In the experiment, we find $K'$ has a significant impact on the quality of rendered images, while the threshold $\eta$ has relatively low impact, but needs to be fit with $K'$. Here we provide default settings that give a satisfying quality with low computation cost: $K' = 20, \eta = 0.01$.

Figure 11 shows the rendered cuboids using different $K'$ and $\eta$. Here the results demonstrate that inadequate $K'$ will lead to some dark region around the boundary of kernels, which we think is caused by the hard cutoff of the boundary. On the other hand, decreasing the threshold $\eta$ could make the object denser (less transparent), but need more kernels (higher $K'$) to avoid the artifacts.

### A.4 MESH & POINT CLOUD CONVERTER

We develop a simple mesh converter, which converts triangular meshes into isotropic Gaussian ellipsoids, and a point cloud converter. In the mesh converter, we retain all original vertices on the mesh and compute the $\mathbf{\Sigma}_k$ using the distance between each vertex and its connected neighbors.

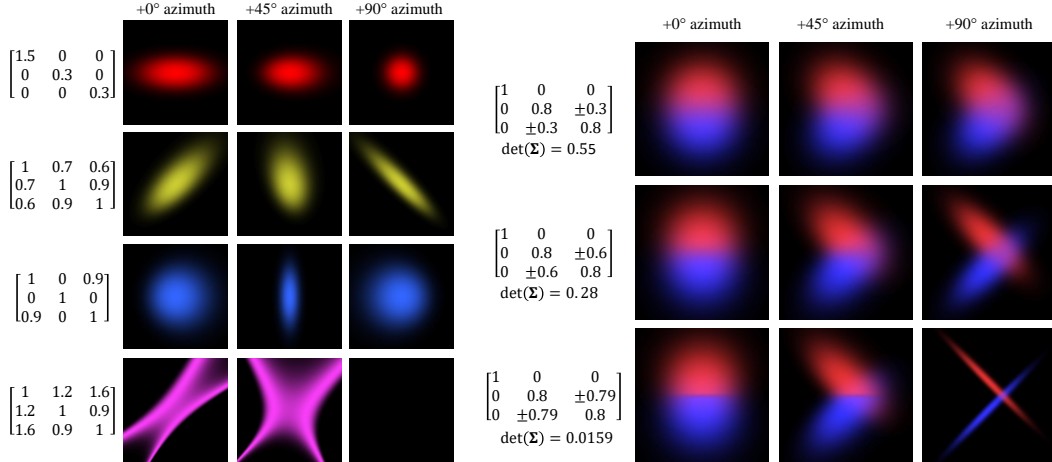

Figure 12: Rendering single anisotropic ellipsoidal Gaussian. Left column shows the $\mathbf{\Sigma}$ for the kernel. We render the kernel under 3 different viewpoint: $0°$ azimuth, $45°$ azimuth, and $90°$ azimuth.

Figure 13: Rendering flattened Gaussian ellipsoids to approximate the 2D Gaussian ellipses. For each row, we show two Gaussian ellipsoids viewed in 3 different viewpoints. From top to bottom: decrease $\det(\mathbf{\Sigma})$ to flatten Gaussian ellipsoids.

Specifically, for each vertex, we compute the average length $d_k$ of edges connected to that vertex. Then $\mathbf{\Sigma}_k$ is computed via:

$$\mathbf{\Sigma}_k = \begin{bmatrix} \sigma_k & 0 & 0 \\ 0 & \sigma_k & 0 \\ 0 & 0 & \sigma_k \end{bmatrix} \tag{29}$$

where $\sigma_k$ is computed via the coverage rate $\zeta$ and $d_k$,

$$\sigma_k = \frac{(d_k/2)^2}{\log(1/\zeta)} \tag{30}$$

Similarly, in the point cloud converter, the $\mathbf{\Sigma}_k$ is controlled with the same function, but the $d_k$ is determined by the distance to $m$ nearest points of the target points.

Since the concurrent mesh converter does not consider the shape of the triangles, admittedly we think this could be improved via converting each triangle into an anisotropic Gaussian ellipsoid, which we are still working on.

## B  ADDITIONAL RENDERING RESULTS

### B.1  RENDERING ANISOTROPIC GAUSSIAN ELLIPSOIDS

As Figure 12 shows, VoGE rendering pipeline natively supports anisotropic ellipsoidal Gaussian kernels, where for each kernel the spatial variance is represented via the $3 \times 3$ symmetric matrix $\mathbf{\Sigma}_k$. Note that, the spatial covariances, *e.g.*, $\sigma_{k,xy}$, cannot exceed square root of dot product of the two variances, *e.g.*, $\sqrt{\sigma_{k,xx}\sigma_{k,yy}}$, otherwise, the kernel will become hyperbola instead of ellipsoids (as the last row in Figure 12 shows).

On the other hand, we suggest that ellipsoidal Gaussian kernels can also **approximate the 2D Gaussian ellipses** (the representation used in DSS Yifan et al. (2019)), which can be simply done by set $\det(\mathbf{\Sigma}) \to 0$, where $\det$ is the determinant of matrix. Figure 13 shows the rendering result using flattened Gaussian ellipsoids. As we demonstrated in the third row, VoGE rendering pipeline allows rendering the surface-liked representations in a stable manner.

### B.2  RENDERING SURFACE NORMAL

As Figure 14 shows, we render CAD models provided by *The Stanford 3D Scanning Repository* Curless & Levoy (1996). Specifically, we use our mesh converter to convert the meshes provide by

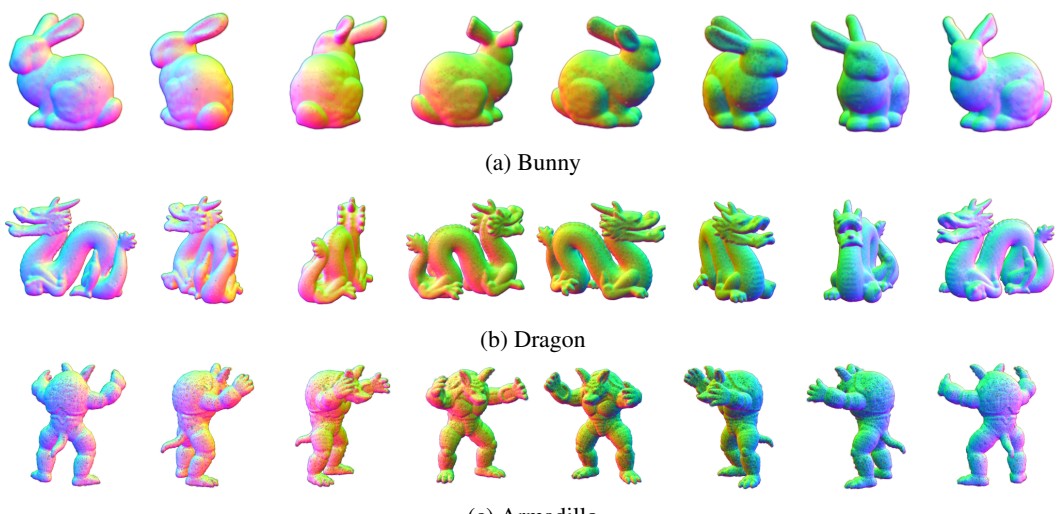

(a) Bunny

(b) Dragon

(c) Armadillo

Figure 14: Rendering the surface normal using VoGE under 8 different viewpoints. The Gaussian ellipsoids are converted from meshes provided by *The Stanford 3D Scanning Repository*.

the dataset into Gaussian ellipsoids. In detail, the Bunny contains 8171 vertices, the Dragon contains 22998 vertices, and the Armadillo contains 33792 vertices. During rendering, we compute surface normals via PyTorch3D and use the normals as the RGB value of each vertex. Then we render the Gaussian Ellipsoids in 8 different viewpoints, and interpolate the RGB value into images.

### B.3 RENDERING QUALITY VS NUMBER OF GAUSSIANS

Figure 16 shows surface normal rendering quality of VoGE using different number of Gaussians. We also include comparison of rendering quality of VoGE vs PyTorch3D mesh renderer. In each image, we control a same number of Gaussians vs mesh vertices, which gives similar number of parameters that $9 * N_{Gauss}$ vs $3 * N_{verts} + 3 * N_{faces}$. Here we observe that increasing number of Gaussians will significant improve rendering quality. Admittedly, VoGE renderer gives slight fuzzier boundary compare to mesh renderer.

### B.4 LIGHTING WITH EXTERNAL NORMALS

Although Gaussian ellipsoids do not contain surface normal information (since they are represented as volume), VoGE still can utilize surface normal via processing them as an extra attribute in an external channel as we describe in section B.2. Once the surface normals are rendered, the light diffusion method in the traditional shader can be used to integrate lighting information into VoGE rendering pipeline. Figure 15 shows the results that integrate lighting information when rendering the Stanford bunny mesh using VoGE. Specifically, we first render the surface normals computed via PyTorch3D into an image-liked map (same as the process in section B.2). Then we use the diffuse function (*PyTorch3D.renderer.lighting*), to compute the brightness of the rendered bunny under a point light. In the visualization, we place the light source at variant locations, while using a fully white texture on the bunny.

### B.5 RENDERING POINT CLOUDS

Figure 17 shows the point clouds rendering results using VoGE and PyTorch3D. We follow the *Render a colored point cloud* from PyTorch3D official tutorial Ravi et al. (2022b). Specifically, we use the PittsburghBridge point cloud provided by PyTorch3D, which contains 438544 points with RGB color for each point respectively. We first convert the point cloud into Gaussian ellipsoids using the method described in A.4. Then we render the Gaussian ellipsoids using the same configuration (Except the camera. As the tutorial uses orthogonal camera, which concurrently we don't support,

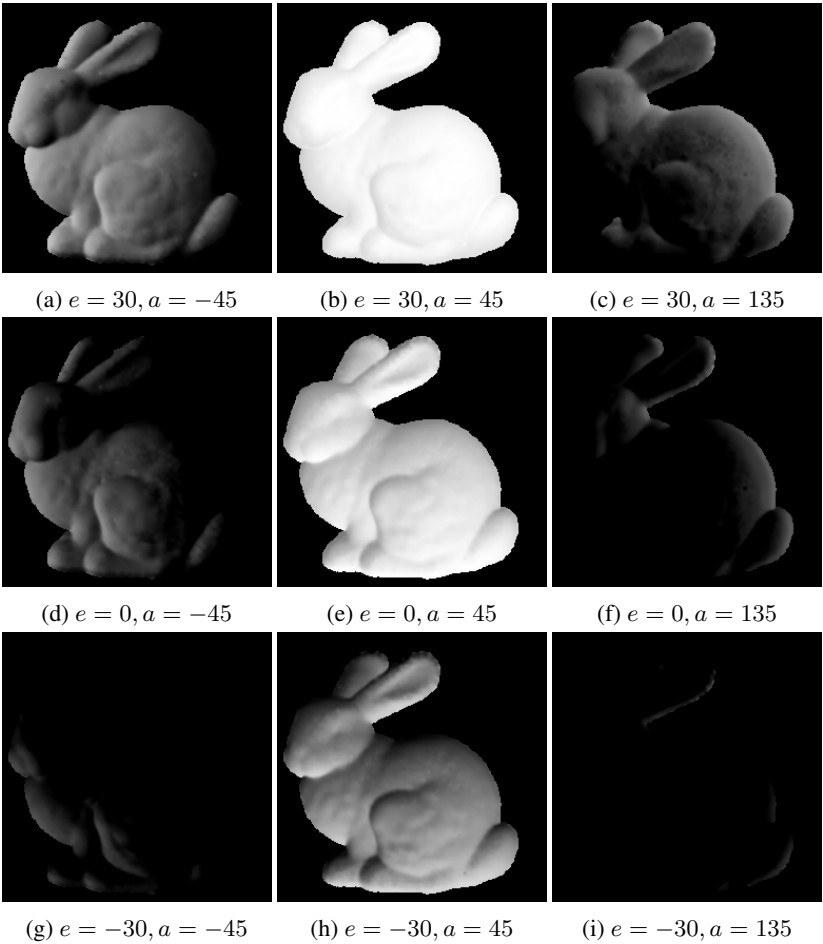

(a) $e = 30, a = -45$    (b) $e = 30, a = 45$    (c) $e = 30, a = 135$

(d) $e = 0, a = -45$    (e) $e = 0, a = 45$    (f) $e = 0, a = 135$

(g) $e = -30, a = -45$    (h) $e = -30, a = 45$    (i) $e = -30, a = 135$

Figure 15: Lighting rendered mesh using external normals. We first render the surface normals of the bunny mesh using VoGE. Then we use the light diffusion functions provided by PyTorch3D to light the render surface normal. For each image, we place a point light source in the object space using different elevations ($e$) and azimuth ($a$). The distance from the light source to the object center is fixed as 1.

Table 4: (Left) Ablation study for object pose estimation on PASCAL3D+. We control the coverage rate $\zeta$ when computing $\mathbf{\Sigma}$, higher $\zeta$ gives larger values in $\mathbf{\Sigma}$. w/o grad $T(\mathbf{r})$ means we block the gradient from $T(\mathbf{r})$, while w/o grad $\rho(\mathbf{r})$ means gradient on $e^{q_k}$ in Equation 8 is blocked.

| Exp. Setup | $ACC_{\frac{\pi}{6}}$ | $ACC_{\frac{\pi}{18}}$ | MedErr |
|---|---|---|---|
| $\zeta = 0.2$ | 89.9 | 68.7 | 7.0 |
| $\zeta = 0.5$ (standard) | 90.1 | 69.2 | 6.9 |
| $\zeta = 0.8$ | 90.3 | 64.7 | 8.5 |
| w/o grad $T(\mathbf{r})$ | 47.1 | 18.9 | 39.1 |
| w/o grad $\rho(\mathbf{r})$ | 31.7 | 8.0 | 48.1 |

we alternate the camera using a PerspectiveCamera with a similar viewing scope). The qualitative results demonstrate VoGE a better quality with smoother boundaries.

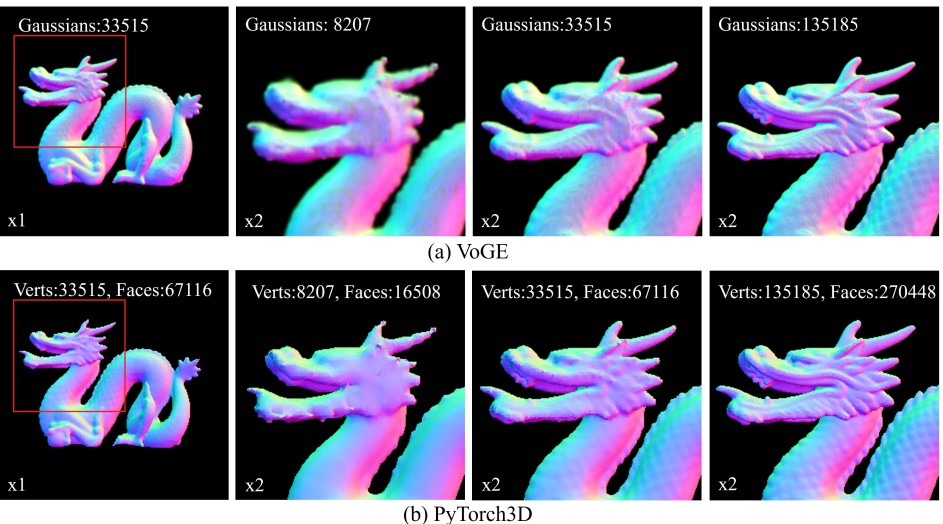

Figure 16: Comparison of rendering quality with number of primitives using VoGE vs PyTorch3D hard mesh renderer.

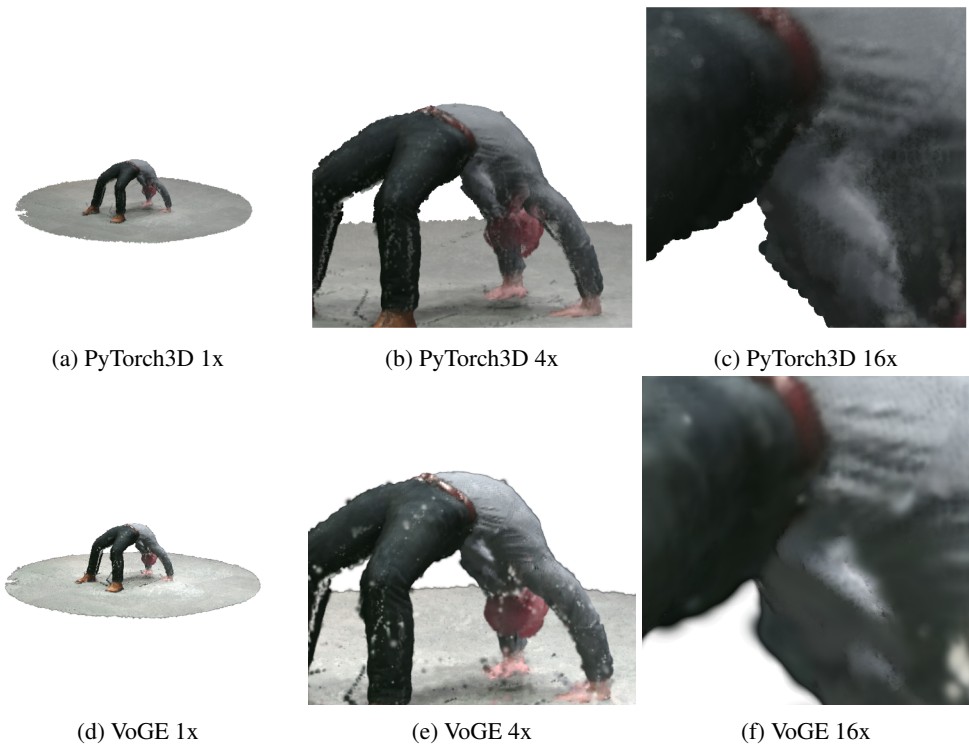

Figure 17: Rendering point clouds using VoGE and PyTorch3D. In the 1x visualization, we render the image with camera with standard focal length. In 4x and 16x we zoom-in the camera by increase the focal length by 4x and 16x.

Table 5: Per category result for in-wild object pose estimation results on PASCAL3D+. Results are reported in Accuracy (percentage, higher better) and Median Error (degree, lower better).

| | | aero | bike | boat | bottle | bus | car | chair | table | mbike | sofa | train | tv | Mean |
|---|---|---|---|---|---|---|---|---|---|---|---|---|---|---|
| $\uparrow ACC_{\frac{\pi}{6}}$ | Res50-General | 83.0 | 79.6 | 73.1 | 87.9 | 96.8 | 95.5 | 91.1 | 82.0 | 80.7 | 97.0 | 94.9 | 83.3 | 88.1 |
| | Res50-Specific | 79.5 | 75.8 | 73.5 | 90.3 | 93.5 | 95.6 | 89.1 | 82.4 | 79.7 | 96.3 | 96.0 | 84.6 | 87.6 |
| | StarMap | 85.5 | **84.4** | 65.0 | **93.0** | 98.0 | 97.8 | **94.4** | 82.7 | 85.3 | **97.5** | 93.8 | **89.4** | 89.4 |
| | NeMo+SoftRas | 80.8 | 79.2 | 70.3 | 88.0 | 89.1 | 98.4 | 85.6 | 74.9 | 82.0 | 95.7 | 76.2 | 82.3 | 85.3 |
| | NeMo+DSS | 77.2 | 69.3 | 65.4 | 83.7 | 91.4 | 96.5 | 80.9 | 67.8 | 71.0 | 89.9 | 76.3 | 77.5 | 81.1 |
| | NeMo+PyTorch3D | 82.2 | 78.4 | 68.1 | 88.0 | 91.7 | 98.2 | 87.0 | 76.9 | 85.0 | 95.0 | 83.0 | 82.2 | 86.1 |
| | NeMo+VoGE(ours) | **89.7** | 82.6 | **77.7** | 88.2 | **98.1** | **99** | 90.5 | **84.8** | **87.5** | 94.9 | 89.2 | 83.9 | **90.1** |
| $\uparrow ACC_{\frac{\pi}{18}}$ | Res50-General | 31.3 | 25.7 | 23.9 | 35.9 | 67.2 | 63.5 | 37.0 | 40.2 | 18.9 | 62.5 | 51.2 | 24.9 | 44.6 |
| | Res50-Specific | 29.1 | 22.9 | 25.3 | 39.0 | 62.7 | 62.9 | 37.5 | 42.0 | 19.5 | 57.5 | 50.2 | 25.4 | 43.9 |
| | StarMap | 49.8 | 34.2 | 25.4 | **56.8** | 90.3 | 81.9 | **67.1** | 57.5 | 27.7 | **70.3** | 69.7 | 40.0 | 59.5 |
| | NeMo+SoftRas | 47.5 | 26.2 | 36.2 | 49.9 | 85.5 | 94.5 | 46.7 | 50.7 | 29.8 | 59.5 | 63.9 | 42.6 | 59.7 |
| | NeMo+DSS | 22.8 | 10.2 | 23.7 | 37.8 | 52.8 | 38.9 | 23.1 | 15.9 | 12.1 | 31.7 | 18.7 | 25.7 | 27.8 |
| | NeMo+PyTorch3D | 49.7 | 29.5 | 37.7 | 49.3 | 89.3 | 94.7 | 49.5 | 52.9 | 29.0 | 58.5 | 70.1 | 42.4 | 61.0 |
| | NeMo+VoGE(ours) | **61.4** | **40.3** | **51.2** | 53.9 | **93.8** | **96.7** | 58.6 | **70.8** | **39.6** | 63.8 | **79.3** | **47.9** | **69.2** |
| $\downarrow MedErr$ | Res50-General | 13.3 | 15.9 | 15.6 | 12.1 | 8.9 | 8.8 | 11.5 | 11.4 | 16.6 | 8.7 | 9.9 | 15.8 | 11.7 |
| | Res50-Specific | 14.2 | 17.3 | 15.4 | 11.7 | 9.0 | 8.8 | 12.0 | 11.0 | 17.1 | 9.2 | 10.0 | 14.9 | 11.8 |
| | StarMap | 10.0 | 14.0 | 19.7 | **8.8** | 3.2 | 4.2 | **6.9** | 8.5 | 14.5 | **6.8** | 6.7 | 12.1 | 9.0 |
| | NeMo+SoftRas | 10.6 | 17.3 | 15.1 | 10.0 | 3.3 | 3.4 | 10.4 | 9.9 | 14.9 | 8.4 | 6.1 | 12.3 | 9.1 |
| | NeMo+DSS | 16.6 | 23.0 | 19.5 | 13.1 | 9.3 | 11.7 | 16.2 | 18.7 | 21.9 | 14.2 | 20.5 | 18.2 | 16.1 |
| | NeMo+PyTorch3D | 10.1 | 16.3 | 14.9 | 10.2 | 3.2 | 3.2 | 10.1 | 9.3 | 14.1 | 8.6 | 5.4 | 12.2 | 8.8 |
| | NeMo+VoGE(ours) | **7.5** | **12.8** | **9.8** | 9.1 | **2.6** | **2.9** | 8.6 | **5.8** | **12.5** | 7.7 | **4.3** | **10.5** | **6.9** |

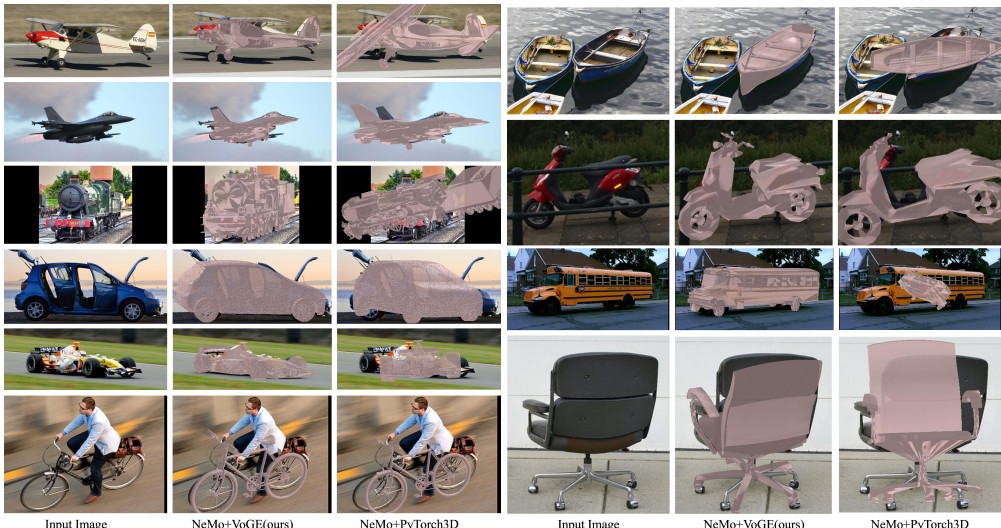

Figure 18: Additional qualitative in-wild object pose estimation results for NeMo+VoGE and NeMo+PyTorch3D.

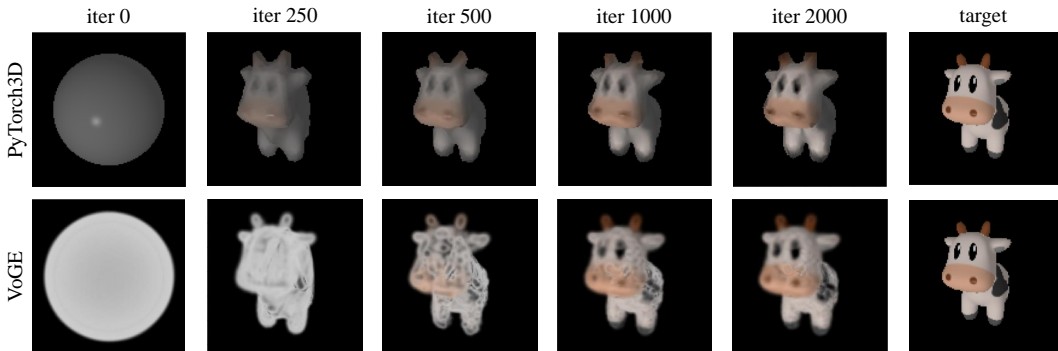

Figure 19: The shape fitting process regarding optimization iterations. We visualize both VoGE and PyTorch3D with all constraints.

## C    ADDITIONAL EXPERIMENT RESULTS

### C.1    IN-WILD OBJECT POSE ESTIMATION

**Ablation Study.** As Table 4 shows, we conduct controlled experiments to validate the effects of different geometric primitives. Using the method we described in 3.2, we develop tools that convert triangle meshes to Gaussian ellipsoids, where a tunable parameter, coverage rate, is used to control the intersection rate between nearby Gaussian ellipsoids. Specifically, the higher coverage rate gives the large $\Sigma$, which makes the feature more smooth but also fuzzy, vice versa. As the results demonstrate, increasing $\Sigma$ can increase the rough performance under $\frac{\pi}{6}$, while reducing it can improve the performance under the more accurate evaluation threshold. We also ablate the affect regarding block part of the gradient in Equation 8. Specifically, we conduct two experiments on all kernels, we block the gradient on $T(l_k)$ and $e^{q_k}$ respectively. The results show blocking either term leads significant negative impact on the final performance.

**Additional Results.** Table 5 shows the per-category object pose estimation results on PASCAL3D+ dataset (L0). All NeMo Wang et al. (2020a) baseline results and ours are conducted using the single cuboid setting described in NeMo. Specifically, Gaussian ellipsoids used in VoGE is converted from the same single cuboid mesh models provided by NeMo (coverage rate $\zeta = 0.5$).

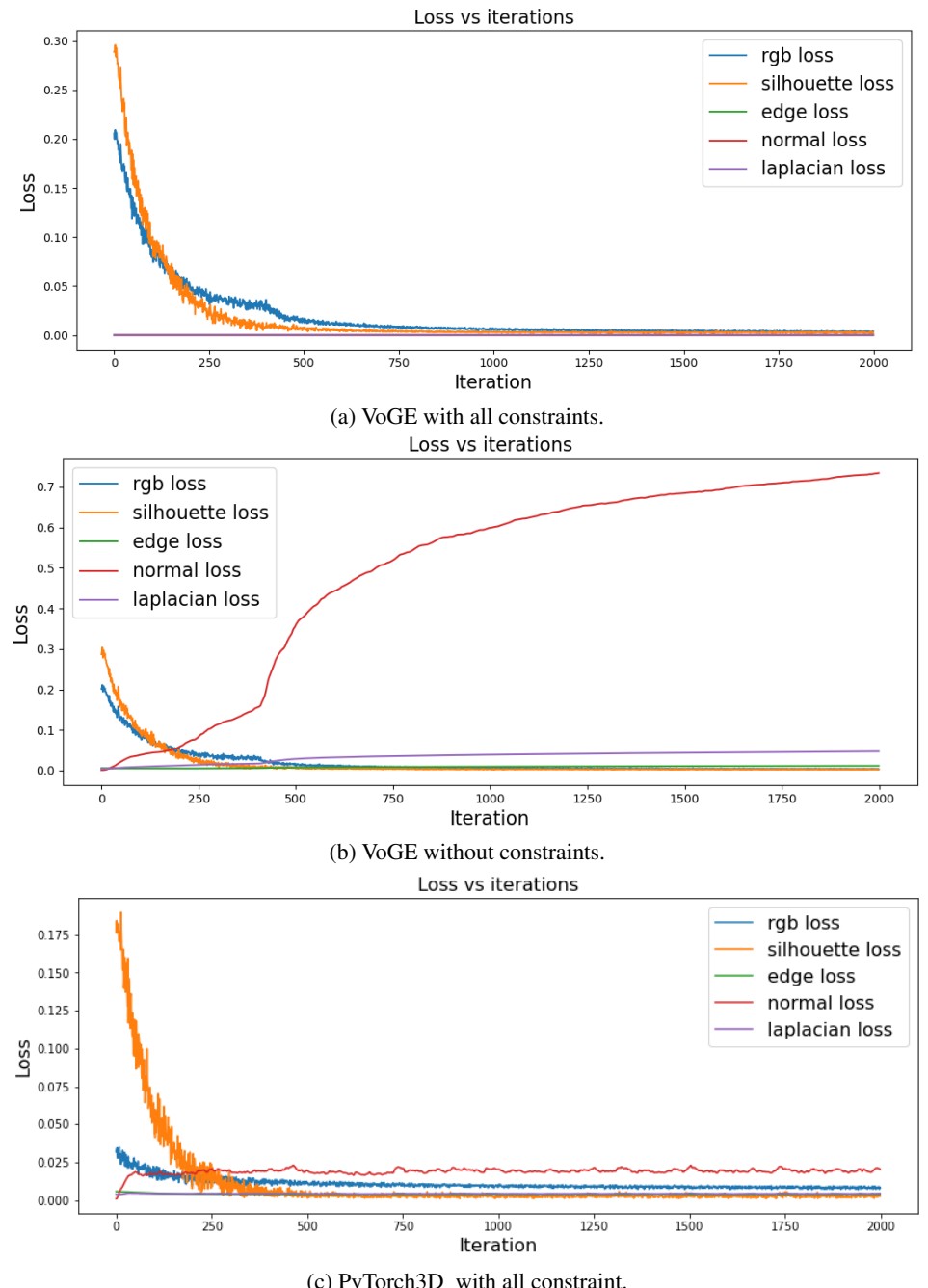

(a) VoGE with all constraints.

(b) VoGE without constraints.

(c) PyTorch3D  with all constraint.

Figure 20: Losses in the shape fitting experiment. Note that for the VoGE without constraint, we only calculate the geometry loss but not compute gradient with those losses.

Figure 18 shows the additional qualitative results of the object pose estimation. In the visualization, we use a standard graphic renderer to render the original CAD models provide by PASCAL3D+ dataset under the predicted pose, and superimpose the rendered object onto the input image.

## C.2 Texture Extraction and Rerendering

Figure 21 shows the additional texture extraction and rerendering results on car, bus and boat images from PASCAL3D+ dataset. Interestingly, Figure 21 (g) shows the texture extraction using VoGE demonstrate stratifying generation ability on those out distributed cases.

## C.3 Shape Fitting via Inverse Rendering

Figure 20 shows the losses in the multi-viewed shape fitting experiment. Specifically, we plot the losses regarding optimization iterations using the method provided by *fit a mesh with texture via rendering* from PyTorch3D official tutorial Ravi et al. (2022a). Note the geometry constraint losses except normal remain relatively low in VoGE without constraints experiment. We think such results demonstrate the optimization process using VoGE can give correct gradient toward the optimal solution effectively, that even without geometry constraint the tightness of the Gaussian ellipsoids is still retained. As for the normal consistency loss, since we use the volume Gaussian ellipsoids, the surface normal directions are no longer informative.

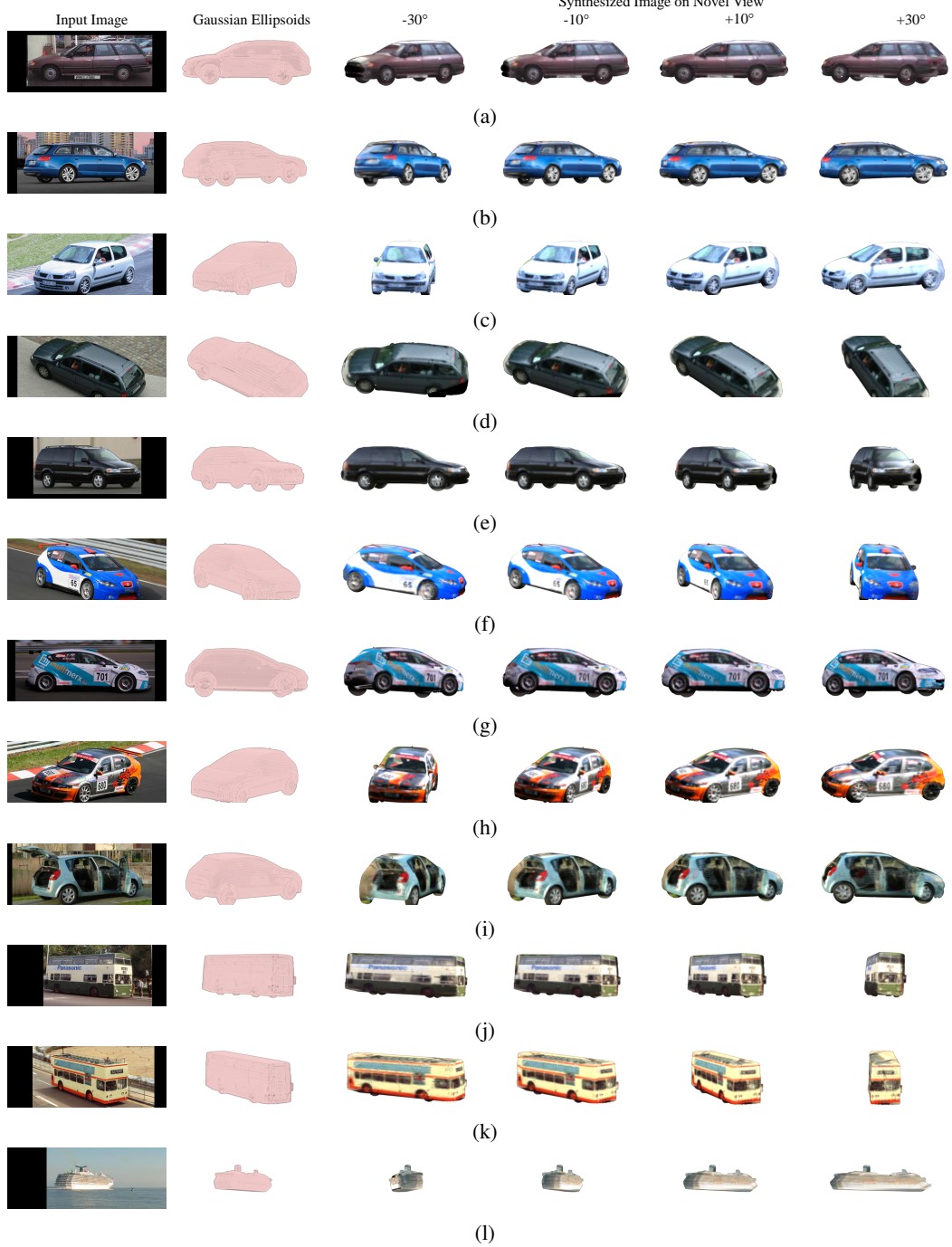

Figure 21: Additional results for texture extraction experiment on car, bus and boat category in PASCAL3D+ dataset. We extract texture using a in-wild image and Gaussian ellipsoids with corresponded viewpoint, and render under novel viewpoint.

