# OpenReview forum: "VoGE: A Differentiable Volume Renderer using Gaussian Ellipsoids for Analysis-by-Synthesis"
_ICLR.cc/2023/Conference — ICLR 2023 poster_

### Official Review · Reviewer_qZYG · 2022-10-23

**Confidence:** 4
**Correctness:** 3
**Technical Novelty And Significance:** 3
**Empirical Novelty And Significance:** Not applicable
**Recommendation:** 5

**Clarity, Quality, Novelty And Reproducibility:**

With respect to clarity, there are a large number of typos and grammatical errors that would need to be corrected. I list a few below:
- Intro: "Liu et al. Liu et al. (2019)"
- Related works: "Blinn Blinn" and other typos for the references. There are similar typos with the references throughout the paper that should be corrected.
- Volume density aggregation: This sentence has many grammatical errors: "However, computing the integral using brute force is so [computationally] inefficient that [it is] infeasible for [current devices]"
- Section 4.4 "official tutorial pyt"

There are a number of other technical questions I had, which I did not see addressed:
- For the texture extraction and rerendering task is the CAD model already aligned to the image? If so, how does this compare to other simple baselines? For example, couldn't one simply map each vertex to a color and render using, e.g., rasterization and barycentric interpolation of the colors on the mesh faces?
- When ray tracing how does one know which ellipsoids contribute to the ray? Does this require a hit test against all ellipsoids?
- How are the ellipsoids initialized when doing shape reconstruction?

As for novelty, there is a long history of using "Gaussian transfer functions" for volume visualizations. Some of the references in this line of work appear in the paper. So, the main novelty here is the incorporation of Gaussian primitives into the differentiable rendering framework.

With respect to reproducibility, I think there is no problem as the authors also promise to release the code.

Additional comment:
- Fig. 2: indicate the row of the image for which the plot is given (e.g., with a dashed line across the image).

**Strength And Weaknesses:**

The idea of using Gaussian ellipsoids in the context of volumetric rendering, while not new, is nevertheless interesting in the context of differentiable rendering because one can compute the rendering integrals analytically. The paper does a good job of conveying the efficiency of the approach, as even megapixel images can be rendered in real time for a scene with thousands of Gaussian ellipsoids.

The quantitative results also shows that the method performs well for the pose estimation task (which involves aligning ground truth geometry to an input image).

Still, the paper has a few weaknesses: there are a large number of typos and grammatical errors that hinder the clarity of the paper, and there are several details lacking from the technical description and evaluation of the method that seem important.
- There are no quantitative results that evaluate the texture extraction and rerendering, occlusion reasoning, and shape fitting via inverse rendering tasks.
- It's not always clear how the method is initialized, which is one potential drawback. How many Gaussian ellipsoids are required in the experiments and how does the rendering quality depend on this? This seems like a critical parameter to evaluate.
- Given the connection to volumetric rendering methods and the interest in multiview reconstruction, I was hoping to see more reconstruction results and how this compares to methods that use a neural network to parameterize the volume density and color. Is there an advantage here? Or does the requirement of initializing a constant number of ellipsoids a priori limit the performance?

**Summary Of The Paper:**

The paper presents a method for differentiable volume rendering using a set of Gaussian ellipsoid primitives. The key idea is to use a large set of Gaussian ellipsoids to parameterize the density and color within a volume and then to render using ray marching to solve the volume rendering equation. Conveniently, integrals along rays through the ellipsoids can be computed in closed form, leading to computationally efficient rendering. The method is demonstrated on a pose estimation task, where a ground truth mesh is aligned to a model, as well as for rendering textures, point clouds, and for shape fitting from multiview imagery. Overall, the method appears to outperform other techniques for the pose estimation task, and qualitative results suggest good performance for other tasks.


**Summary Of The Review:**

Overall, while I think the idea of using Gaussian ellipsoids as a primitive for differentiable rendering is interesting and potentially very reasonable (as it builds on a long legacy of using Gaussian primitives for volume rendering), I have some reservations about the evaluation and lack of quantitative comparisons for 3/4 of the presented tasks. I find it difficult to get a sense of how well the method performs. There are also some technical details which appear to be missing from the paper. As it stands, I lean slightly negative on the paper.

---

> ### Author Response · Authors · 2022-11-18
> **Reply to Reviewer LqFK**
>
> ### There are no quantitative results that evaluate texture extraction and rerendering, occlusion reasoning, and shape fitting via inverse rendering tasks.
>
> Here we evaluate the average PSNR over all 20 viewing directions in the inverse rendering task, higher better:
>
> | Method |  VoGE  |  PyTorch3D  |
> | ------ | ------ | ----------- |
> | PSNR   | 24.63  |  18.41      |
>
> The distance error (L2 distance between the groundturth and prediction) for the occlusion reasoning experiment, lower better:
>
> | Method  |  VoGE  |  PyTorch3D  |
> | ------- | ------ | ----------- |
> | Cuboid1 | 0.0096 |  6.1847     |
> | Cuboid2 | 0.0089 |  0.6962     |
>
>
> ### It's not always clear how the method is initialized, which is one potential drawback. How many Gaussian ellipsoids are required in the experiments and how does the rendering quality depend on this? This seems like a critical parameter to evaluate.
>
> The rendering quality does depend on the number of ellipsoids, we include a qualitative illustration and include it in Appendix Figure 16. We also conduct a quantitative experiment for image qualities using a different number of Gaussian ellipsoids, please refer to the reply to reviewer cWiy.
>
> ### For the texture extraction and the rerendering task is the CAD model already aligned to the image? If so, how does this compare to other simple baselines? For example, couldn't one simply map each vertex to a color and render using, e.g., rasterization and barycentric interpolation of the colors on the mesh faces?
> Yes, the mesh is aligned with the object in the image. We include the baseline approach and comparison as the reviewer suggests. The results are included in the **common questions** section of this rebuttal. Specifically, we rasterize the mesh and use the barycentric coordinates to compute a weighted sum for colors near the projected location of each vertex. Then, we render the mesh with vertex color into an RGB image under a novel view. The number of vertices in the mesh is the same as the number of ellipsoids in our approach.
>
>
> ### When ray tracing how does one know which ellipsoids contribute to the ray? Does this require a hit test against all ellipsoids?
> In our implementation, we first conduct the coarse rasterization process (following [this implementation](https://github.com/facebookresearch/pytorch3d/blob/main/pytorch3d/csrc/rasterize_coarse/rasterize_coarse.cu)), which selects a subset of ellipsoids (~10% of all ellipsoids) that potentially interacted with the ray. Then, we do the hit test against this subset of ellipsoids.
>
> ### How are the ellipsoids initialized when doing shape reconstruction?
> The ellipsoids are initialized to be uniformly distributed on the surface of the geodesic sphere with 2562 ellipsoids, where the distance between the centers of each two adjacent ellipsoids is the same. The sigmas are initialized as the same value along all directions using the mesh to VoGE convertor.

---

> ### Author Response · Authors · 2022-12-13
> **Reply to Reviewer LqFK**
>
> Thank you once again for your reviews and suggestions. At the end of the discussion stage, we would like to make sure that the updated information of this paper (the VoGE differentiable rendering pipeline using a volumetric Guassian-based representation that achieves high fidelity and real-time rendering speed) has been brought to your attention. We hope these additional updates & revisions and comments from other reviewers can help to solve your concerns on this work.

---

### Official Review · Reviewer_LqFK · 2022-10-25

**Confidence:** 4
**Clarity, Quality, Novelty And Reproducibility:** 1. The paper is math-intensive, espec…
**Correctness:** 3
**Technical Novelty And Significance:** 3
**Empirical Novelty And Significance:** Not applicable
**Recommendation:** 8

**Strength And Weaknesses:**

Strengths
1. Compared to rasterization-based differentiable renderers, the proposed method
utilizes volume density and can better reason about occlusions and
overlapped components.

2. Compared to implicit volumetric representation, the usage of Gaussian
ellipsoid is more intuitive and can be easily converted from other
representations such as meshes and point clouds.

3. The paper derives an efficient approximation for the aggregation function
that is differentiable with respect to both visible and invisible components.

4. The paper shows that the proposed renderer achieves better performance than
baseline methods on multiple downstream tasks.

Weakness
1. When talking about implicit representations such as Mildenhall2020, the paper
says that it lacks modifiability and interpretability. It's not clear to me why
the proposed Gaussian ellipsoids representation can be better in these aspects,
and it's worth adding more discussions here. Also recently there are new
volumetric representations such as Plenoxels and Direct Voxel Grid Optimization
that apply feature grids + MLPs. It will also be better to discuss these related works,
and how the proposed method will be better.

2. It's also worth discussing relations to physically-based differentiable
rendering methods, such as
* Differentiable Monte Carlo Ray Tracing through Edge Sampling
* Differentiable Signed Distance Function Rendering
* A Differential Theory of Radiative Transfer

3. In the inverse rendering tasks, the produced results are blurry. What are the
limiting factors for the proposed method to generate sharper renderings? Can it
be improved by increasing the number of Gaussian ellipsoids?
Also, the paper says a silhouette is used. I am wondering whether it's really
necessary, considering that volume-based methods such as Mildenhall2020 do not
require mask supervision. Similarly, it's also not clear to me why additional
constraints are needed.

4. In Figures 14 and 16, it seems that the method produced normal maps with
obvious artifacts, especially on the Armadillo and the dragon scene. What are
the reasons for those artifacts?

5. I think one advantage of the proposed method over rasterization-based
method is that it's using volume rendering and therefore can handle translucent
objects. It will be interesting to add a comparison on tasks like fitting a
translucent object.

**Summary Of The Paper:**

The paper proposes a novel differentiable renderer. It represents the scene
using a set of Gaussian ellipsoids that determines the density of each point.
During rendering, it traces each ray and accumulates the contribution of
each ellipsoid using volume rendering. The paper demonstrates the effectiveness
of the renderer on tasks such as pose estimation and shape fitting, and shows
that it achieves better performance that rasterization-based differentiable
renderer such as Pytorch3D.

**Summary Of The Review:**

Overall, I believe that the combination of 3D Gaussian and volume renderings is
novel and technically sound. The paper shows the effectiveness of the proposed
method on multiple tasks. Therefore, I hold a positive attitude toward the
paper.

---

> ### Author Response · Authors · 2022-11-18
> **Reply to Reviewer LqFK**
>
>
> ### Related works
> Thanks for the suggestions for the related works, we have included all of them in our revision.
>
> ### In the inverse rendering tasks, the produced results are blurry. What are the limiting factors for the proposed method to generate sharper renderings? Can it be improved by increasing the number of Gaussian ellipsoids? Also, the paper says a silhouette is used. I am wondering whether it's really necessary, considering that volume-based methods such as Mildenhall2020 do not require mask supervision. Similarly, it's also not clear to me why additional constraints are needed.
>
> The main limiting factor for the image quality in this experiment is the number of ellipsoids used. For a fair comparison, we use the same number of Gaussians as the number of vertices in the PyTorch3D mesh. We show that by using more Gaussians the problem with the blurriness is reduced.
> For mask supervision, we suggest that the silhouette loss is useful for learning object geometry[1] for both volume-based and surface-based rendering methods. In the learning process of NeRF, the optimization is applied to the volume densities local. Whereas, in the optimization process of VoGE, the gradients apply to the location and shape of the Gaussians.
>
> [1] Wang, P., Liu, L., Liu, Y., Theobalt, C., Komura, T., & Wang, W. (2021). Neus: Learning neural implicit surfaces by volume rendering for multi-view reconstruction. arXiv preprint arXiv:2106.10689.
>
> ### In Figures 14 and 16, it seems that the method produced normal maps with obvious artifacts, especially on the Armadillo and the dragon scene. What are the reasons for those artifacts?
>
> The artifacts are mainly introduced when we preprocess the meshes. Specifically, in order to convert the Armadillo mesh into Gaussians, we first use the “remesh” operator in Blender to uniformly sample the mesh vertices, while retaining the topology of the object. However, this process can introduce the small artifacts on the surface of the object, e.g., pits and bulges.
>
>
> ### I think one advantage of the proposed method over the rasterization-based method is that it's using volume rendering and therefore can handle translucent objects. It will be interesting to add a comparison on tasks like fitting a translucent object.
>
> We very much agree with the reviewer regarding the point that VoGE is better for modeling transparent objects in compared to the rasterization-based base method. However, an issue with real-world transparent objects is that rays will refract when passing through the object (otherwise, the object will be only a colored mask and not visible as an object). However, the refraction are not handled by our current implementation (neither does the NeRF model), unless some special techniques are introduced to take into account refraction, which is beyond the scope of this work.
>
> ### The reproducibility remains a concern, considering that there are many components in the paper with subtle details. It will be great if the authors can release the code.
> We promise to release the code with detailed instructions about installation, documentation, and demos. We have packed VoGE into easily used PyTorch APIs and hope the VoGE renderer can be useful for the community.

---

> ### Author Response · Authors · 2022-12-13
> **Reply to Reviewer LqFK**
>
> Thank you once again for your reviews and suggestions. At the end of the discussion stage, we would like to make sure that the updated information of this paper (the VoGE differentiable rendering pipeline using a volumetric Guassian-based representation that achieves high fidelity and real-time rendering speed) has been brought to your attention. We hope these additional updates & revisions and comments from other reviewers can help to solve your concerns on this work.

---

### Official Review · Reviewer_cWiy · 2022-10-25

**Confidence:** 4
**Correctness:** 2
**Technical Novelty And Significance:** 2
**Empirical Novelty And Significance:** 2
**Recommendation:** 3

**Clarity, Quality, Novelty And Reproducibility:**

There are issues with the clarity of the writing, see my comments in the previous section.

A few additional references should be added,

"Learning Deformable Tetrahedral Meshes for 3D Reconstruction"
NeurIPS 2020

"Differentiable Monte Carlo ray tracing through edge sampling"
ACM Transactions on Graphics 2 TOG 2018

"Mitsuba 2: A Retargetable Forward and Inverse Renderer"
SIGGRAPH Asia 2019

"Modular Primitives for High-Performance Differentiable Rendering"
ACM Transactions on Graphics 2020


**Strength And Weaknesses:**

With 3D gaussians, the paper introduces an interesting representation for differentiable volumetric rendering. The algorithm for rendering a set of gaussians is sound and the experiments demonstrate strong results on the pose estimation task and plausible behavior on the inverse rendering experiments.

---

The paper is unfortunately quite difficult to follow. The formatting suggests that not much time was spent on proof-reading the paper, as the citation format expected by the authors is inconsistent with the ICLR format. Some sections are tiring to read from the frequent duplication of author names in the citations (see Section 2, first paragraph).

The authors use non-standard terminology for common concepts, which does not help the readability. For example, in Table 1, it would be better to use conventional graphics terms like these:

 - "Component" -> "Geometric primitive"
 - "Component Tracking" -> "Visibility algorithm"
 - "Aggregation" -> "Blending"

Certain statements are confusing and likely incorrect, for example the statement that "Graphics renderers use explicit object representations, which represent objects as a set of isotropic components."  (Sec. 2) (What components are referred to? What is "isotropic" about conventional graphics rendering?)

---

While the paper is quite clear in how a set of gaussians is rendered, the way in which shapes are converted to sets of gaussians is not sufficiently discussed. There is one paragraph on the bottom of page 4 that mentions how to pick the covariance matrix, but it is not mentioned in the main text how the number or position of the gaussians is chosen. From the appendix, it becomes clear that there is a 1-to-1 mapping with variance based on the neighboring edge length.

There are potential issues with this approach, in that the softness of the rendering (both spatially and along the ray) will depend on the mesh tesselation. Coarse meshes will be very soft, while finely tesselated meshes will converge to point-based splatting. The paper should include an analysis of this effect with an ablation and/or failure cases.

---

The mentioned coarse-to-fine approximations are not well motivated and their impact not thoroughly evaluated.

---

Section 4.2 / Figure 7:

If the CAD model and pose is provided, this result is expected of any renderer that supports RGB textures. This result does not require a differentiable renderer, as each vertex (or gaussian) can simply be projected into the image, where its RGB value can be looked up without back propagation or optimization. It is not clear why training or symmetry information (Figure caption) would help.

---

Figure 8: How many gaussians are rendered for each cuboid? It appears that there are more than 8 gaussians, but the appendix mentions a 1-to-1 vertex to gaussian mapping. Can each gaussian vary its position independently, or are the gaussian positions constrained by the cuboid shape?


**Summary Of The Paper:**

The paper introduces a differentiable rendering algoithm that converts 3D meshes into a collection of 3D gaussians and ray traces the resulting scene. The conversions to volumetric gaussians smoothes the visibility function of the scene, similar to the smooth visibility and blending functions of soft differentiable rasterizers (SoftRasterizer, PyTorch3D).

The paper describes how to project a set of 3D gaussians along a 1D camera ray, and how to solve the resulting volumetric rendering integral in closed form.

Two approximations are mentioned: One sub-samples the geometry by only rendering 10% of all gaussians in a coarse-to-fine approach. The other optimization clusters nearby gaussians, which further reduces the number of 3D density samples.

The paper describes how arbitrary attributes can be associated with the 3D gaussians, which allows for the rendering of RGB textures or neural features.

The paper presents several experiments, including pose estimation of CAD shapes, texturing, shape fitting, and a synthetic occlusion experiment.


**Summary Of The Review:**

The closed form solution for rendering a set of gaussians is a worthwhile contribution, but overall the exposition in the paper is quite difficult to follow. Experiment 4.1 appears to be a strong result, but Experiment 4.2 is in my opinion not very relevant. Experiments 4.3 and 4.4 are quite basic and would benefit from a more thorough discussion of how mesh conversion and coarse-to-fine rendering is handled.

---

> ### Author Response · Authors · 2022-11-18
> **Reply to Reviewer cWiy**
>
> ### The authors use non-standard terminology for common concepts, which does not help the readability.
> We have revised our paper according to your suggestion.
>
> ### The way in which shapes are converted to sets of Gaussians is not sufficiently discussed.
> We have included a detailed description in Section 3.3 in the revision.
>
> ### Coarse meshes will be very soft, while finely tesselated meshes will converge to point-based splatting. The paper should include an analysis of this effect with an ablation and/or failure cases.
>
> Here we include a quantitative evaluation of the rendering quality regarding the number of Gaussians in the object. Specifically, we render the Stanford bunny. First, we use a high-resolution object to render an image (indicated with the *), which contains 738493 Gaussians. Then we evaluate the PSNR for the rendered image when downgrade the number of Gaussians.
>
> | Number of Gaussians |  738493 |  184225  |  45855  | 11413 | 2780 | 685 |
> | ------------------- | ------- |-------|-------|-------|-------|-------|
> | PSNR                | inf *   | 37.09 | 31.16 | 26.68 | 22.87 | 18.57 |
>
> ### Figure 8: How many gaussians are rendered for each cuboid? It appears that there are more than 8 gaussians, but the appendix mentions a 1-to-1 vertex to gaussian mapping. Can each gaussian vary its position independently, or are the gaussian positions constrained by the cuboid shape?
>
> The cuboid contains 4000 Gaussians that are distributed uniformly on the surface of the cuboid (the distance between centers of each nearest pair of Gaussians are same). For a fair comparison, the Cuboid mesh for PyTorch3D contains the same number of vertices at the same locations as the centers of Gaussians. In this experiment, the shape of each cuboid is rigid, which means the centers of kernels in each cuboid are fixed.
> In the following, we also show an example if we do not constrain the shape to be rigid.
> In this interesting example, you will see how the Gaussian kernels interact with each other, and at one stage they are connected to rebuild the cuboid.
> [Video Example Without Shape Constrain](https://drive.google.com/file/d/1dJGXVdA90WKU8YTSfNLOlOh0UfdUliQq/view?usp=sharing), [Video Example With Shape Constrain](https://drive.google.com/file/d/1EG8WGr22CY2ipfY7VQV3iPwNoJsiukvn/view?usp=sharing).
>
> ### Section 4.2 / Figure 7 are in my opinion not very relevant.
> We will move this section into the appendix and add the ablation experiment described in the rebuttal to our paper in the final revision. For the explanation of this experiment, please refer to our answers to the questions that are common for all reviewers.

---

> ### Author Response · Authors · 2022-11-18
> **Reply to Reviewer cWiy**
>
> ### The mentioned coarse-to-fine approximations are not well motivated and their impact not thoroughly evaluated.
>
> The motivation for introducing the coarse-to-fine strategy is to reduce the memory needed and increase the rendering speed.
> Here we include a quantitative evaluation of rendering speed and rendering quality when using different coarse-to-fine strategies, i.e. if we use the coarse rasterizer and the K nearest ellipsoids.
> To evaluate this, we use CUDA-VoGE to render the Stanford bunny, which includes 8171 Gaussians.
> The naïve implementation in the first row does not use the coarse rasterizer. We report the rendering time with only the forward process and a whole forward-backward process.
> We compute a loss between the rendered image with a black image to calculate the gradient on the center of the Gaussians.
> Also, we compute a PSNR to evaluate the rendering quality. Specifically, the PSNR is computed between the image rendered via 160 nearest Gaussians without coarse rasterizer (indicated with * ) and each other rendered image.
> Additionally, we also evaluate the quality of the gradient via computing an L1 distance between the gradients under setting 100 nearest Gaussians without coarse rasterizer (indicated with * ) and each other case.
> The NA indicates we cannot run this experiment due to the limitation of memory of our GPU.
>
>
> | Coarse Rasterizer |           | Naïve    | Coarse   | Naïve    | Coarse   | Naïve   | Coarse  | Naïve   | Coarse  | Naïve  | Coarse | Naïve   | Coarse  | Naïve    | Coarse   |
> |--------------------|-----------|----------|----------|----------|----------|---------|---------|---------|---------|--------|--------|---------|---------|----------|----------|
> | K nearest Gaussians  |           | 160      | 160      | 100      | 100      | 60      | 60      | 40      | 40      | 20     | 20     | 10      | 10      | 5        | 5        |
> | Forward            | Time (ms)     | 73.60    | 66.84    | 45.09    | 40.69    | 32.19   | 27.78   | 28.42   | 25.18   | 21.91  | 19.78  | 21.77   | 13.35   | 24.77    | 22.76    |
> |                    | Mem (MB)      | 19417.88 | 19417.88 | 7707.88  | 7642.88  | 2792.88 | 2792.88 | 1267.88 | 1267.88 | 342.88 | 342.88 | 326.63  | 326.63  | 326.63   | 326.63   |
> | Forward + Backward | Time (ms)     | NA       | NA       | 77.78    | 73.08    | 43.55   | 43.72   | 24.10   | 23.10   | 19.85  | 20.99  | 21.12   | 16.06   | 15.73    | 17.19    |
> |                    | Mem  (MB) | NA       | NA       | 20229.16 | 20230.16 | 7339.16 | 7339.16 | 3294.16 | 3294.16 | 849.16 | 849.16 | 326.63  | 326.63  | 326.63   | 326.63   |
> |                    | PSNR      | Inf *    | 99.91    | 154.47   | 99.91    | 94.24   | 93.20   | 66.07   | 66.07   | 41.94  | 41.94  | 26.44   | 26.44   | 18.49    | 18.49    |
> |                    | Grad_Diff | NA       | NA       | 0 *   | 0.0036   | 0.0110  | 0.0084  | 0.2753  | 0.2728  | 8.3679 | 8.3655 | 43.4750 | 43.4720 | 147.2485 | 147.2524 |
>
> ### A few additional references should be added
> Thanks for the suggestion regarding references, we have included all of them in our revision.

---

> ### Author Response · Authors · 2022-12-13
> **Reply to Reviewer cWiy**
>
> Thank you once again for your reviews and suggestions. At the end of the discussion stage, we would like to make sure that the updated information of this paper (the VoGE differentiable rendering pipeline using a volumetric Guassian-based representation that achieves high fidelity and real-time rendering speed) has been brought to your attention. We hope these additional updates & revisions and comments from other reviewers can help to solve your concerns on this work.

---

### Official Review · Reviewer_ppJY · 2022-11-01

**Confidence:** 5
**Correctness:** 3
**Technical Novelty And Significance:** 3
**Empirical Novelty And Significance:** 2
**Recommendation:** 5

**Clarity, Quality, Novelty And Reproducibility:**

Quality: I think this is good work, and does contribute a nice idea to the field of differentiable/inverse rendering. I do take small issues with some of the results and figures. For example, I don't find Figure 5 very illuminating without also comparing rendering fidelity. That is, I don't care if one method is faster than the other if has worse fidelity and vice versa. Also, I am a little curious about the novel view synthesis results of Figure 7-- without training, the network cannot predict any new information-- so is this result simply due to the bilateral symmetry of a car and ellipsoidal coloring that can be rotated geometrically? In general, I would want more explanation given for this surprising result.

Clarity: The clarity, at least for me (and I do research in this are), was very bad. Yet, I believe there is a much more straightforward exposition possible, I just don't think the authors took the time to structure their exposition for comprehension. I wish the paper was better written because I really do like the idea and execution, but I don't think it is publication-ready at the current time. For example, I would put the method for turning a mesh into Gaussians in the main text, and I would also explain how the K nearest ellipsoids are selected for ray tracing (as well as compare compute times & fidelity with and without this technique). Other time I think a simple proofread could have helped clarity-- for example, in Section 3.4, the word "donates" is used several times, but I think the word should have been "denotes"?

Originality: I think the idea is original and I think the author's well-covered earlier work that has similar ideas. I also appreciate Table 1 for contextualizing the method.

**Strength And Weaknesses:**

Strengths: The idea is good and the background is well established and covered. As a differentiable rendering method, the idea to decompose the scene into "ellipsoidal blobs" allows much better occlusion/de-occlusion reasoning for the applications covered in the paper, and therefore, better performance than image space differentiable rendering techniques.

Weaknesses: The paper is fairly confusing, and the exposition needs to be reworked. I had trouble understanding it during the first read-through, in part due to Equation references that do not appear in the main text (is the main paper referencing equations from Appendices?). There is also a lot of typographical errors, including simple citation errors that could have been easily spotted and fixed with a single read-through. This tells me that, despite its length and the richness of the idea, this paper was surely rushed.



**Summary Of The Paper:**

This paper presents a new differentiable volumetric rendering method based on ray casting through a discrete number of Gaussian shaped occlusion density functions. In practice, assumptions are made to simplify the volumetric rendering equation in terms of peak/max densities in order to reduce the rendering equation to something that can be computed efficiently in closed form. The authors show competitive speed to PyTorch3D, with applications to pose estimation, novel view synthesis, and inverse rendering.

**Summary Of The Review:**

I like this work but do not like this paper. I think the work should be published and would be of considerate interest to the community. However, I think the paper is not well-organized and the ideas are not well-presented. Part of this problem is due to simple typographical issues, but part of this problem is due to hard to understand and haphazard exposition.

---

> ### Author Response · Authors · 2022-11-18
> **Reply to Reviewer ppJY**
>
> ### Rendering fidelity regrading Figure 5
> We are not sure about what the reviewer means here by rendering fidelity, since there is no ground-truth image to compare the rendering to. For each data point in the plot in Figure 5, both the number of primitives and image resolutions are controlled to be the same. For (a), the number of primitives per pixel is 20 for both methods, while in the (b) plot, the image size is always set to 256. We hope that this clarifies.
>
> ### Novel view synthesis results of Figure 7
> We include a more detailed discussion, please refer to the **Common Questions**.
>
> ### I would also explain how the K nearest ellipsoids are selected for ray tracing (as well as compare compute times & fidelity with and without this technique).
> Thanks for this suggestion. We include a **more detailed ablation** regarding K nearest ellipsoids and coarse rasterization in reply to reviewer cWiy.
> Also, the evaluation of the rendering speed when using different K of nearest ellipsoids is illustrated in Figure 5 (b) in the paper.
>
> ### The clarity was very bad
> We have revised our paper to improve the clarity following all your suggestions and those of other reviewers, including terminology, and clarification of the motivation. Please refer to our revised submission.

---

> ### Author Response · Authors · 2022-12-13
> **Reply to Reviewer ppJY**
>
> Thank you once again for your reviews and suggestions. At the end of the discussion stage, we would like to make sure that the updated information of this paper (the VoGE differentiable rendering pipeline using a volumetric Guassian-based representation that achieves high fidelity and real-time rendering speed) has been brought to your attention. We hope these additional updates \& revisions and comments from other reviewers can help to solve your concerns on this work.

---

### Author Response · Authors · 2022-11-18
**To all reviewers**

First, we thank all reviewers for their efforts in reading and reviewing our submission. We also thank the reviewers for the detailed comments, questions, and suggestions.

We appreciate that all reviewers acknowledge the major scientific contribution of our proposed differentiable rendering pipeline, which leverages a volumetric Guassian-based representation to achieve high fidelity and real-time rendering speed.

## Common Questions
### Exposition issue

We thank all reviewers for giving detailed feedback on the exposition and typographical errors. We have proofread and revised the article in depth. We also updated the abstract and introduction following the reviewers' suggestions to improve the clarity. Please refer to the **revised version of our paper**.

### Adding the conversion from mesh to Gaussian Ellipsoids to main text

We have now included the details about the conversion from mesh to Gaussians from the appendix in the main paper. Please refer to Section 3.3 in our revised paper for details.

### Novel view synthesis results of Figure 7
The goal for this texture extraction experiment is to visually demonstrate the feature/texture extraction ability as described in Section 3.4. This operation can be useful in vision tasks like 3D reconstruction. However, we agree that this is not a very critical experiment, and we plan to move it to the appendix in the final revision, to add more details into the main paper recarding the conversion from meshes into Gaussian ellipsoids (see previous point).

Nevertheless, we want to give more details here regarding the experiment in Figure 7, since we think it demonstrates a promising property of our proposed Gaussian ellipsoid representation. To illustrate the difference between VoGE and previous approaches, we include two different baseline approaches for comparison. First, we sample vertex colors via projecting each vertex onto the image and measuring the color for the nearest pixel (no matter whether it is occluded or not), then we render the result with the PyTorch3D renderer.
The other baseline uses rasterization and barycentric interpolation of the colors on the mesh faces (as suggested by Reviewer qZYG) using the soft renderer. Specifically, for each pixel, we collect 25 nearest faces and use the distance-based blending method to compute overall weight, then we render the collected vertex colors using the same soft renderer (the variance parameters in the soft renderer are set to 1e-3, and gamma is 1e-4).
The results: [VoGE](https://drive.google.com/file/d/1vu23pel3wo8yFjgmkx8rFEa4Iy1Hc2FS/view?usp=share_link), [Barycentric](https://drive.google.com/file/d/1NSa8fFo17OOfp9DruhahnzZ3-vSA60VG/view?usp=share_link), [Nearest Pixel](https://drive.google.com/file/d/1O5izGVEhOgnI3ntJbtUzzVDX4UYwnsu3/view?usp=share_link).

---

### Decision · Program_Chairs · 2023-01-20

**Decision:**

Accept: poster

**Justification For Why Not Higher Score:**

There is not a wide consensus among the reviewers in favour of this work, and there were some flaws in the initial version of this work.

**Justification For Why Not Lower Score:**

See meta review above.

**Metareview: Summary, Strengths And Weaknesses:**

This work focuses on a novel proposed method for differentiable volume rendering.

The reviewers pointed out, the main strengths of this work are the performance of the proposed method, especially its performance on some downstream tasks compared to existing baselines. They also commanded the general proposed idea, and how well the literature is covered. They found that the main weakness was the exposition, both in terms of clarity and use of unconventional terms.

These have been appropriately addressed by the authors during the rebuttal process.



**Note From Pc:**

if the above contains the word "oral" or "spotlight" please see: "oral" presentation means -> notable-top-5% and "spotlight" means -> notable-top-25%. As stated in our emails, we are disassociating presentation type from AC recommendations